# IMPROVING MOLECULE-LANGUAGE ALIGNMENT WITH HIERARCHICAL GRAPH TOKENIZATION

## ABSTRACT

Recently there has been a surge of interest in extending the success of large language models (LLMs) to graph modality, such as molecules. As LLMs are predominantly trained with 1D text data, most existing approaches adopt a graph neural network to represent a molecule as a series of node tokens and feed these tokens to LLMs for molecule-language alignment. Despite achieving some successes, existing approaches have overlooked the hierarchical structures that are inherent in molecules. Specifically, in molecular graphs, the high-order structural information contains rich semantics of molecular functional groups, which encode crucial biochemical functionalities of the molecules. We establish a simple benchmark showing that neglecting the hierarchical information in graph tokenization will lead to subpar molecule-language alignment and severe hallucination in generated outputs. To address this problem, we propose a novel strategy called **HI**erarchical **G**rap**H** **T**okenization (`HIGHT`). `HIGHT` employs a hierarchical graph tokenizer that extracts and encodes the hierarchy of node, motif, and graph levels of informative tokens to improve the graph perception of LLMs. `HIGHT` also adopts an augmented molecule-language supervised fine-tuning dataset, enriched with the hierarchical graph information, to further enhance the molecule-language alignment. Extensive experiments on $14$ molecule-centric benchmarks confirm the effectiveness of `HIGHT` in reducing hallucination by $40\%$, as well as significant improvements in various molecule-language downstream tasks.

## 1 INTRODUCTION

Large language models (LLMs) have demonstrated impressive capabilities in understanding and processing natural languages (Radford et al., 2019; OpenAI, 2022; Touvron et al., 2023a; Bubeck et al., 2023). Recently, there has been a surge of interest in extending the capabilities of LLMs to graph modality (Jin et al., 2023; Li et al., 2023d; Wei et al., 2024; Mao et al., 2024; Fan et al., 2024) such as social networks (Tang et al., 2023; Chen et al., 2024) and molecular graphs (Liu et al., 2023d; Zhao et al., 2023; Cao et al., 2023; Li et al., 2024). Inspired by the success of large vision-language models (Zhang et al., 2024; Zhu et al., 2023; Liu et al., 2023a), existing large graph-language models (LGLMs) predominantly adopt a graph neural network (GNN) (Kipf & Welling, 2017; Hamilton et al., 2017; Xu et al., 2019) to tokenize graph information as a series of node embeddings (or node tokens), and then leverage an adapter such as a Multi-layer perceptron (MLP) or a Q-former (Li et al., 2023b) to transform the node tokens into those compatible with LLMs (Fan et al., 2024). To facilitate the alignment of graph and language modalities, LGLMs will undergo a graph-language instruction tuning stage with the graph and the corresponding caption data, so as to realize the graph-language alignment (Jin et al., 2023; Li et al., 2023d; Fan et al., 2024).

Despite achieving certain success, the graph tokenization in existing LGLMs neglects the *essential hierarchical structures* that are inherent in graph data (Ying et al., 2018). Especially, in molecular graphs, the high-order structural information, such as motifs or functional groups, contains rich semantics of the biochemical functionalities of the molecules (Milo et al., 2002; Bohacek et al., 1996; Sterling & Irwin, 2015). For example, the existence of the hydroxide functional group ("-OH") in small molecules often indicates a higher water solubility. Therefore, the perception of functional groups in a molecule is essential for LLMs to understand the molecules. Intuitively, feeding LLMs with only node tokens makes the molecule understanding harder as LLMs have to learn to combine atoms in a functional group during the instruction tuning phase. However, atoms are usually treated

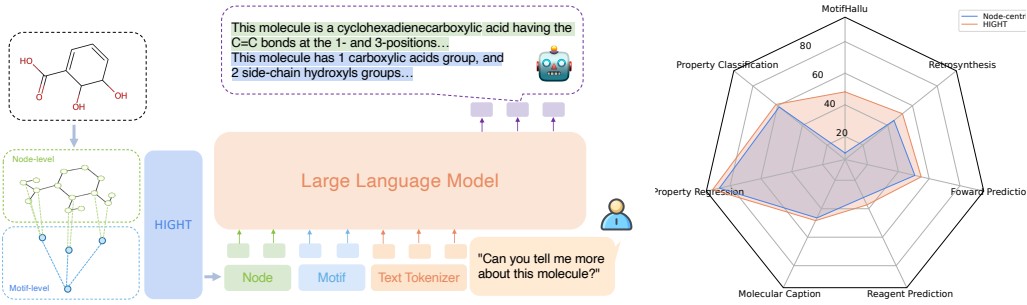

(a) Overview of the HIGHT framework.  (b) Summary of performance.

Figure 1: Illustration of HIGHT. *(a)* Given a molecule (i.e., PubChem ID 3, *5,6-Dihydroxycyclohexa-1,3-diene-1-carboxylic acid*), `HIGHT` detects the motifs and incorporates the "supernodes" for each motif (The whole graph is also considered as a "super motif".). Then, `HIGHT` tokenizes the molecule into both node-level (i.e., atoms) and motif-level (i.e., functional groups) tokens. The hierarchical view enables LLMs to align the molecular structures and the language descriptions of the molecule better. *(b)* Therefore, `HIGHT` significantly reduces the hallucination of LGLMs and improves the downstream performance across various molecule-centric tasks. All metrics are transformed a bit such that a higher number means a better downstream task performance.

as separate tokens in LGLMs and there is often a lack of supervision signal to prompt about the combinations of specific motifs. Consequently, neglecting the hierarchical information will lead to subpar graph-language alignment and severe hallucination. To demonstrate the issue, we construct a simple benchmark called `MotifHallu` that asks LLMs about the existence of common functional groups. Surprisingly, we find that existing LGLMs consistently answer "Yes" for any functional groups, as demonstrated in Sec. 3.2. It then raises a challenging research question:

*Is there a way to incorporate the intrinsic hierarchical graph information into LLMs?*

In this paper, we study the problem with a focus on molecular data, and introduce a new graph-language alignment strategy called **HI**erarchical **G**rap**H T**okenization (`HIGHT`). As shown in Fig. 1, `HIGHT` includes a hierarchical graph tokenizer, as well as a hierarchical molecular instruction tuning dataset to facilitate a better alignment of molecule and language modalities. Inspired by the success of hierarchical GNNs in molecular representation learning (ZHANG et al., 2021; Zang et al., 2023; Inae et al., 2023; Luong & Singh, 2023), we transform the original molecular graph into a hierarchical graph with motif and graph nodes added in. Then, we employ a Vector Quantized-Variational AutoEncoder (VQVAE) to obtain atom-level, motif-level, and graph-level tokens separately with the self-supervised tasks (Zang et al., 2023). To retain more original structural information, we further attach Laplacian positional encodings to the tokens. After that, we adopt a multi-level adapter consisting of three adapters processing atom-level, motif-level, and graph-level tokens, respectively, before feeding them into the LLMs. In addition, to facilitate the use of hierarchical information encoded by the tokens, we augment the original molecular instruction tuning dataset with motif descriptions. Our contributions can be summarized as follows:

- To the best of our knowledge, we are the first to propose incorporating the hierarchical graph information into LGLMs with new architectures and instruction tuning dataset `HiPubChem`.

- To facilitate the graph-language alignment study on molecular graphs, we also propose the first hallucination benchmark `MotifHallu` based on the existence of common functional groups.

- We conduct extensive experiments with 14 real-world molecular and reaction comprehension benchmarks. The results show that `HIGHT` significantly reduces the hallucination on `MotifHallu` by up to 40% and consistently improves the downstream molecule-language performances.

## 2 PRELIMINARIES

We begin by introducing preliminary concepts and related works of LGLMs.

**Large Graph-Language Models.** As LLMs have demonstrated great capabilities across a wide range of natural language tasks, there has been an increasing interest in extending LLMs to broader applications where the text data are associated with the structure information (i.e., graphs) (Jin et al., 2023; Li et al., 2023d; Wei et al., 2024; Mao et al., 2024; Fan et al., 2024). A graph can be denoted as $\mathcal{G} = (\mathcal{V}, \mathcal{E})$ with a set of $n$ nodes $v \in \mathcal{V}$ and a set of $m$ edges $(u, v) \in \mathcal{E}$. Each node $u$ has node attributes as $\boldsymbol{x}_u \in \mathbb{R}^d$ and each edge $(u, v)$ has edge attributes $e_{u,v} \in \mathbb{R}^{d_e}$. A number of LGLMs have been developed to process graph-text associated dataset $\mathcal{D} = \{\mathcal{G}, \boldsymbol{c}\}$, where $\boldsymbol{c} = [c_1, ..., c_{l_c}]$ refers to the caption of the graph $\mathcal{G}$. For node-centric tasks, $\boldsymbol{c}_i$ will associate with the nodes (Tang et al., 2023), while in this paper we focus on graph-centric tasks, i.e., molecules and molecular captions (Liu et al., 2023d). Usually, an $l$-layer GNN is employed to encode a graph as:

$$\boldsymbol{h}_u^{(l)} = \text{COMBINE}(\boldsymbol{h}_u^{(l-1)}, \text{AGG}(\{(\boldsymbol{h}_u^{(l-1)}, \boldsymbol{h}_v^{(l-1)}, e_{uv})|v \in \mathcal{N}(u)\})), \quad (1)$$

where $\boldsymbol{h}_u^{(l)} \in \mathbb{R}^h$ refers to the node embedding of node $u$ after $l$ layers of GNN, $\text{AGG}(\cdot)$ is the aggregation function (e.g., mean) among the information from neighbors of node $u$, and COMBINE is the operator for combining information of node $u$ with its neighbors $\mathcal{N}(u)$ (e.g., concatenation). Then, after $l$ message passing iterations, the graph-level embedding can be obtained as:

$$\boldsymbol{h}_\mathcal{G} = \text{READOUT}\left(\{h_u^{(l)}|u \in \mathcal{V}\}\right), \quad (2)$$

where $\text{READOUT}(\cdot)$ is a pooling operator (e.g., mean pooling) among all the node embeddings. With the representations of the nodes and graphs, LGLMs can fuse the graph and language information in various ways, such as transforming into natural languages describing the graphs (Fatemi et al., 2024), or neural prompts within the LLMs (Tian et al., 2024). In addition, the embeddings can also be leveraged to postprocess the LLM outputs (Liu et al., 2024a). Orthogonal to different fusion mechanisms, in this work, we will focus on transforming graph embeddings into input tokens of LLMs to demonstrate the benefits of hierarchical graph modeling, which can be formulated as (Tang et al., 2023; Chen et al., 2024; Liu et al., 2023d; Zhao et al., 2023; Cao et al., 2023; Li et al., 2024):

$$p_\theta(\boldsymbol{a}|\boldsymbol{q}, \boldsymbol{h}) = \Pi_{i=1}^{l_a} p_\theta(\boldsymbol{a}_i|\boldsymbol{q}, f_n(\boldsymbol{h}), \boldsymbol{a}_{<i}), \quad (3)$$

where the LGLM is required to approximate $p_\theta$ to output the desired answer $\boldsymbol{a}$ given the question $\boldsymbol{q}$, and the graph tokens $\boldsymbol{h}$ adapted with adapter $f_n : \mathbb{R}^h \to \mathbb{R}^{h_e}$ that projects the graph tokens to the embedding space of LLMs. In addition, one could also incorporate the 1D sequence of molecules such as SMILES (Weininger, 1988) into $\boldsymbol{q}$ and $\boldsymbol{a}$ to facilitate the alignment.

**Molecular Foundation Models.** More specifically, this work focuses on one of the most popular graph-language alignment tasks, i.e., molecule-language alignment (Liu et al., 2024c; Pei et al., 2024). In fact, there is a separate line of works aiming to develop language models for molecules and proteins – the language of lives, from 1D sequences such as SMILES (Irwin et al., 2022), 2D molecular graphs (Wang et al., 2022), 3D geometric conformations (Liu et al., 2022; Zhou et al., 2023), to scientific text (Beltagy et al., 2019) and multimodal molecule-text data (Liu et al., 2023b; Luo et al., 2023a; Christofidellis et al., 2023; Liu et al., 2024b; Su et al., 2022; Zeng et al., 2022). The adopted backbones range from encoder-decoder architectures such as MolT5 (Edwards et al., 2022) and Galactica (Taylor et al., 2022), to auto-regressive language modeling (Luo et al., 2023b; Liu et al., 2023c). Inspired by the success of large vision-language models (Li et al., 2023b; Zhu et al., 2023; Liu et al., 2023a), the community further seeks for developing molecular foundation models built upon existing molecular language models with more sophisticated graph information fusion modules. For example, Liu et al. (2023d); Zhao et al. (2023) develop advanced cross-modal adapters and generalized position embeddings to promote a better graph-language alignment of encoder-decoder based molecular foundation models. Liang et al. (2023); Cao et al. (2023); Li et al. (2024) develop cross-modal adapters for decoder only language models such as Llama (Touvron et al., 2023a). Orthogonal to the aforementioned works, we focus more on what information one shall extract from the graph for better graph-language alignment. We choose to build our methods upon decoder only language models, with the hope to build a versatile agent that can perceive graph information beyond the language, image, and audio modalities (Xi et al., 2023).

**Hierarchical Graph Representation Learning.** The hierarchical nature has been widely and explicitly incorporated in learning high-quality graph representations (Ying et al., 2018). Especially in molecular graphs, the high-order structural information naturally captures the existence of motifs and functional groups. Therefore, the hierarchy of node-motif-graph has been widely applied in

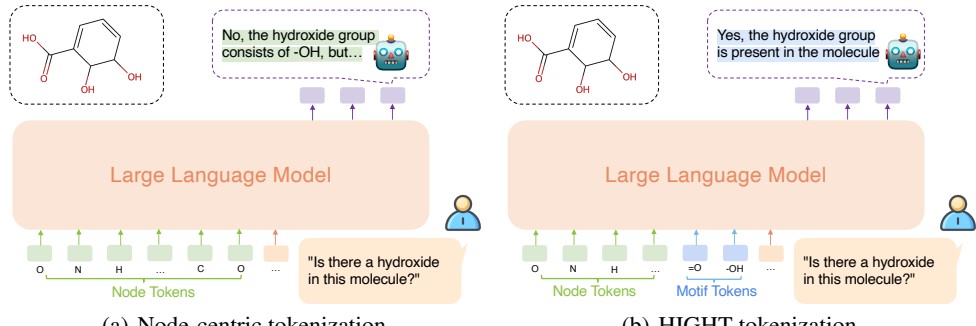

(a) Node-centric tokenization.    (b) HIGHT tokenization.

Figure 2: Illustration of hallucination caused by node-centric tokenization. With only node-level tokens (i.e., discrete atom embeddings), LLMs have to identify and connect the nodes within a specific functional group in order to align useful molecular structures in a molecule to the corresponding language descriptions. Yet, due to the arbitrary order of atoms and position biases in LLMs, it is harder to distinguish each functional group, which further leads to hallucination and subpar alignment.

self-supervised molecular representation learning (ZHANG et al., 2021; Zang et al., 2023; Inae et al., 2023; Luong & Singh, 2023). Nevertheless, it remains unclear how to properly incorporate the hierarchical graph information in graph instruction tuning with LLMs.

## 3 GRAPH TOKENIZATION FOR GRAPH-LANGUAGE ALIGNMENT

Existing LGLMs predominantly tokenize a graph into a series of node embeddings (or node tokens). Despite achieving some success, we find that node-centric tokenization can hardly express motif-level information and will lead to issues such as hallucination as demonstrated in this section.

### 3.1 NODE-CENTRIC TOKENIZATION

Specifically, most existing LGLMs directly take the node tokens from GNNs as inputs to LLMs (Cao et al., 2023):

$$p_\theta(\boldsymbol{a}|\boldsymbol{q}, \boldsymbol{h}) = \Pi_{i=1}^{l_a} p_\theta(\boldsymbol{a}_i|\boldsymbol{q}, f_n(\boldsymbol{h}_1), ..., f_n(\boldsymbol{h}_n), \boldsymbol{a}_{<i}), \quad (4)$$

where $\boldsymbol{h}_1, ..., \boldsymbol{h}_n$ are node embeddings from a GNN typically pretrained through self-supervised learning on large-scale molecular datasets such as ZINC250k (Sterling & Irwin, 2015), $f_n$ is the corresponding adapter to align the node tokens to the LLM tokens. There are various options for tokenizing a molecule, given that a simple supervised trained GNN could produce meaningful tokens (Liu et al., 2023e). In this work, we consider a state-of-the-art node-centric tokenizer from Mole-BERT (Xia et al., 2023) that pretrains a VQVAE (van den Oord et al., 2017) with masked atoms modeling. It constructs a codebook $\mathcal{Z}$ to discretize atoms:

$$z_u = \arg\min_i ||\boldsymbol{h}_u - \boldsymbol{e}_i||_2, \quad (5)$$

where $z_u \in \mathcal{Z}$ is the quantized index of atom $u$, and $\boldsymbol{e}_v$ is the codebook embedding of the $i$-th entry. The codebook is trained through a reconstruction loss with respect to some attribute $\boldsymbol{v}_i$ of atom $i$:

$$\mathcal{L}_r = \frac{1}{n}\sum_{i=1}^{n}(1 - \frac{\boldsymbol{v}_i^T \hat{\boldsymbol{v}}_i}{||\boldsymbol{v}_i|| \cdot ||\hat{\boldsymbol{v}}_i||})^\gamma + \frac{1}{n}\sum_{i=1}^{n}||\text{sg}[\boldsymbol{h}_i] - \boldsymbol{e}_{z_i}||_2^2 + \frac{\beta}{2}\sum_{i=1}^{n}||\text{sg}[\boldsymbol{e}_{z_i}] - \boldsymbol{h}_i||_2^2, \quad (6)$$

where sg[·] is the stop-gradient operator in straight-through estimator (Bengio et al., 2013), $\hat{\boldsymbol{v}}_i$ is the reconstructed attribute of atom $i$ with a decoder, and $\beta$ is a hyperparamter. In Mole-BERT, the attribute is simply the type of atom masked during training. Mole-BERT also manually partitions the codebook into groups of common atoms such as carbon, nitrogen, and oxygen in order to avoid codebook conflicts (Xia et al., 2023).

Intuitively, the trained atom tokens encode some contextual information, such as the neighbors of the atoms. However, node-centric tokenization makes the molecule-language alignment more

challenging, as LLMs have to additionally find the specific nodes to align the corresponding texts during the instruction tuning process. It often encounters the *underspecification* issue during the alignment. For example, in molecules, motifs or functional groups usually capture rich semantics, and often share many common atoms such as carbon, nitrogen, and oxygen (Bohacek et al., 1996). As shown in Fig. 2, both the carboxylic acid ("R-COOH") and the hydroperoxide ("R-OOH") functional groups all contain two oxygen atoms and a hydrogen atom. For a molecule with hydroperoxide attached to a scaffold with carbon atoms, it would be hard for LLMs to distinguish which functional group is present in the molecule. Furthermore, due to the loss of positional information in the node-centric tokenization (Liang et al., 2023; Cao et al., 2023), the limited expressivity of GNNs (Xu et al., 2019) and the positional biases of auto-regressive LLMs (Lu et al., 2022), it is more challenging for the inner LLM to relate the node-level information within a motif, which will lead to more serious performance degeneration of the graph-language alignment.

### 3.2 MOTIF HALLUCINATION

To understand the issue of node-centric tokenization more clearly, we construct a simple benchmark called `MotifHallu`, which measures the hallucination of common functional groups by LGLMs. Specifically, we consider the 38 common functional groups in RDKit[1] and leverage RDKit (Landrum, 2016) to detect the existence of the functional groups within a molecule. We leverage $3,300$ molecules from `ChEBI-20` test split (Edwards et al., 2021), and adopt the query style as for large vision-language models (Li et al., 2023c), which queries the existence of the specific functional group in the molecule:

```
Is there a <functional group name> in the molecule?
```

Then, we detect whether the LGLM gives outputs meaning "Yes" or "No" following the practice in (Li et al., 2023c). For each molecule, we construct questions with positive answers for all kinds of functional groups detected in the molecule, and questions with negative answers for randomly sampled 6 functional groups from the 38 common functional groups in RDKit. Therefore, `MotifHallu` consists of $23,924$ query answer pairs. While it is easy to scale up `MotifHallu` by automatically considering more molecules and a broader scope of functional groups, we find that the current scale is already sufficient to demonstrate the hallucination phenomena in LGLMs.

## 4 HIERARCHICAL GRAPH TOKENIZATION

To mitigate the aforementioned issue, we propose a new strategy called **HI**erarchical **G**rap**H** **T**okenization (`HIGHT`), which contains a hierarchical graph tokenizer that augments the input graph modality, as well as a hierarchical molecular instruction tuning dataset that augments the input language modality, to facilitate the alignment of molecule and language modalities.

### 4.1 HIERARCHICAL GRAPH TOKENIZER

Inspired by the success of hierarchical GNNs in self-supervised molecular representation learning (ZHANG et al., 2021; Zang et al., 2023), we transform the original molecular graph $\mathcal{G}$ into a hierarchical graph $\mathcal{G}'$ with motif and graph nodes added in. Specifically, we leverage the Breaking of Retrosynthetically Interesting Chemical Substructures (BRICS) algorithm (Degen et al., 2008) to detect a set of $k$ motifs in $\mathcal{G}$, denoted as $\mathcal{M} = \{\mathcal{M}^{(1)}, ..., \mathcal{M}^{(k)}, \mathcal{M}^{(k+1)}\}$, where $\mathcal{M}^{(k+1)} = \mathcal{G}$ is the original molecule, without loss of generality. Furthermore, we denote the set of nodes and edges in $\mathcal{M}^{(i)}$ as $\mathcal{V}_m^{(i)}$ and $\mathcal{E}_m^{(i)}$, respectively. Then, we augment the original molecular graph $\mathcal{G}$ as $\mathcal{G}'$ with augmented nodes $\mathcal{V}'$ and edges $\mathcal{E}'$:

$$\mathcal{V}' = \mathcal{V} \cup \{v_m^{(1)}, ..., v_m^{(k+1)}\}, \ \mathcal{E}' = \mathcal{E} \cup (\cup_{i=1}^{k+1} \mathcal{E}_{ma}^{(i)}), \tag{7}$$

where $v_m^{(i)}$ is the motif super nodes added to the original molecule, and $\mathcal{E}_{ma}^{(i)} = \cup_{u \in \mathcal{V}_m^{(i)}} \{(u, v_m^{(i)})\}$ are the augmented edges connecting to the motif super node from nodes within the corresponding motif. We employ separate VQVAEs for atoms and motifs to learn meaningful code embeddings

---

[1]https://github.com/rdkit/rdkit/blob/master/Data/FunctionalGroups.txt

with several self-supervised learning tasks. The reconstructed attributes in Eq. 4 include atom types at the atom-level and the number of atoms at the motif-level, etc., following (Zang et al., 2023).

Meanwhile, merely feeding the motif tokens with node tokens to LLMs still can not help distinguish the motifs from nodes properly. Therefore, we propose to further attach positional encodings $p$ to all of the tokens. We choose to use Laplacian positional embeddings (Dwivedi et al., 2020) while one could easily extend it with other variants (Ying et al., 2021). Since motif (and graph) tokens pose different semantic meanings from atom tokens, we adopt separate adapters for different types of tokens. Denote the motif tokens as $h_m^{(i)}$ for motif $\mathcal{M}^{(i)}$, generation with `HIGHT` tokenizer is as:

$$p_\theta(\boldsymbol{a}|\boldsymbol{q}, \boldsymbol{h}, \boldsymbol{h}_m) = \prod_{i=1}^{l_a} p_\theta(\boldsymbol{a}_i|\boldsymbol{q}, f_n(\boldsymbol{h}_1), ..., f_n(\boldsymbol{h}_n),$$
$$f_m(\boldsymbol{h}_m^{(1)}), ..., f_m(\boldsymbol{h}_m^{(k)}), f_g(\boldsymbol{h}_m^{(k+1)}), \boldsymbol{a}_{<i}), \tag{8}$$

where $f_m(\cdot)$ and $f_g(\cdot)$ are the adapters for BRICS motifs and the original molecules, respectively.

## 4.2 HIERARCHICAL GRAPH INSTRUCTION TUNING DATASET

Although `HIGHT` tokenizer properly extracts the hierarchical information from the input graph modality, it remains challenging to properly align the language information to the corresponding graph information, without the appearance of the respective captions in the texts. For example, if the caption does not contain any information about the water solubility of the hydroxide functional group ("-OH"), LGLMs will never know that "-OH" motif corresponds to the water solubility of the molecule, despite that `HIGHT` tokenizer extracts the "-OH" token. In fact, the commonly used molecular instruction tuning curated from PubChem (Kim et al., 2022) in existing LGLMs (Liu et al., 2023d; Cao et al., 2023; Li et al., 2024), contains surprisingly little information about motifs. Some samples are given in Appendix B.2.

To this end, we propose `HiPubChem`, which augments the molecular instruction tuning dataset with captions of the functional groups. We consider both the positive and negative appearances of motifs when augmenting the instructions. For the positive case, we directly append the caption of all functional groups detected with RDKit:

```
This molecule has <#> of <functional group name> groups.
```

where `<#>` refers to the detected number of the functional group in the molecule, and `<functional group name>` refers to the name of the functional group as listed in Appendix B.2. In addition, we also include a brief introduction of the corresponding functional groups to provide fine-grained information for molecule-language alignment. For the negative case, we randomly sample $k_{\text{neg}}$ that do not appear in the molecule:

```
This molecule has no <functional group name> groups.
```

Despite the simple augmentation strategy, we find that `HiPubChem` significantly reduces the hallucination issue and improves the molecule-language alignment performance.

## 4.3 HIERARCHICAL GRAPH INSTRUCTION TUNING

For the training of `HIGHT`, we use a two-stage instruction tuning as (Cao et al., 2023).

**Stage 1 Alignment Pretraining.** We curate a new molecule-text paired dataset from PubChem following the pipeline of (Liu et al., 2023b). We set the cutoff date by Jan. 2024, and filter out unmatched pairs and low-quality data, which results in 295k molecule-text pairs. Furthermore, we construct the `HiPubChem`-295k dataset based on the curated PubChem-295k dataset. The alignment pretraining stage mainly warms up the adapter to properly project the graph tokens with the LLM embedding space. To avoid feature distortion, both the LLM and the GNN encoder are frozen.

**Stage 2 Task-specific Instruction Tunning.** With a properly trained adapter, we further leverage the task-specific instruction tuning datasets from `MoleculeNet` (Wu et al., 2017), `ChEBI-20` (Mendez et al., 2019), and `Mol-Instructions` (Fang et al., 2024). More details of the instruction tuning

datasets are given in Appendix B. In Stage 2, we still keep the GNN encoder frozen, while tuning both the adapter and the LLM. The LLM is tuned using low-rank adaptation (i.e., LoRA) (Hu et al., 2022) following the common practice.

# 5 EXPERIMENTAL EVALUATION

We conduct extensive experiments to evaluate HIGHT, comparing with previous node-centric tokenization, across 14 real-world tasks including property prediction, molecular description, and chemical reaction prediction. We briefly introduce the setups, and leave the details in Appendix C.

## 5.1 EXPERIMENTAL SETTINGS

We follow the common practice (Cao et al., 2023; Fang et al., 2024) to conduct our experiments.

**Architecture.** The GNN backbone used for producing graph tokens is a 5-layer GIN (Xu et al., 2019) with a hidden dimension of 300. The adapter is implemented as a single-layer MLP. The base LLM adopts the `vicuna-v-1.3-7B` (Chiang et al., 2023). The scale of parameters is around 6.9B.

**Baselines.** We incorporate both the specialist molecular foundation/pretrained models, as well as LLM-based generalist models. The specialist models include expert models pretrained on large-scale molecular datasets and then finetuned on task-specific datasets such as KV-PLM (Zeng et al., 2022), GraphCL (You et al., 2020) and GraphMVP (Liu et al., 2022). The specialist models also include molecule-specialized foundation models that are trained with tremendous molecule-centric datasets such as MolT5-based methods (Edwards et al., 2022), Galactica (Taylor et al., 2022), MoMu (Su et al., 2022), MolFM (Luo et al., 2023a), Uni-Mol (Zhou et al., 2023), MolXPT (Liu et al., 2023c), GIT-Mol (Liu et al., 2024b), and BioMedGPT (Luo et al., 2023b). We adopt the results from previous works (Fang et al., 2024; Cao et al., 2023) when available.

For LLM-based generalist models, we consider LLMs such as ChatGPT (OpenAI, 2022), Llama (Touvron et al., 2023a) as well as instruction tuned LLMs such as Alpaca (Dubois et al., 2023), Baize (Xu et al., 2023), ChatGLM (Zeng et al., 2023) and Vicuna (Chiang et al., 2023). We also consider parameter-efficient finetuned LLMs using the backbone of llama2 (Touvron et al., 2023b) as done by `Mol-Instructions` (Fang et al., 2024). For the node-centric based tokenization, we implement the baseline mainly based on InstructMol (Cao et al., 2023) with a VQVAE tokenizer from `Mole-BERT` (Xia et al., 2023). HIGHT is implemented based on the same architecture with only the tokenizer replaced. We use the suffix "-G" to refer to LLMs with only 2D graph input while using "-GS" to refer to LLMs with both 2D graph and 1D selfies input (Krenn et al., 2019; Fang et al., 2024; Cao et al., 2023). We do not include the baselines with "-GS" for tasks other than `MotifHallu` as we find that incorporating the 1D input does not always bring improvements in the experiments.

**Training and evaluation.** We apply the same optimization protocol to tune LGLMs with node-centric and HIGHT tokenizers for fair comparisons. We train both models with stage 1 by 5 epochs and stage 2 by 5 to 50 epochs as recommended by (Cao et al., 2023).

## 5.2 MOTIF HALLUCINATION

We first evaluate motif hallucination results of the LGLMs with node-centric and with HIGHT tokenization with `MotifHallu`. All the evaluated models only undergo the stage 1 instruction tuning to ensure a fair comparison. We have not included the other generalist baselines as we find they consistently answer "Yes". In addition, in order to avoid the drawbacks that LGLMs may output answers that

| METHOD | F1 (pos) ↑ | F1 (neg) ↑ | Acc ↑ | Yes Ratio |
|---|---|---|---|---|
| *Node-centric Tokenization* | | | | |
| InstructMol-G | 95.7 | 9.5 | 19.9 | 94.5 |
| InstructMol-GS | 97.1 | 10.6 | 20.9 | 94.4 |
| *Hierarchical Tokenization* | | | | |
| **HIGHT-G** | 85.5 | **48.2** | **39.1** | **74.7** |
| **HIGHT-GS** | 84.5 | **42.7** | **35.1** | **73.1** |
| *Ablation variants of HIGHT* | | | | |
| **HIGHT-G w/o HiPubChem** | **96.6** | 12.5 | 21.6 | 96.6 |
| **HIGHT-GS w/o HiPubChem** | **98.2** | 6.5 | 19.4 | 93.3 |

Table 1: Results of motif hallucinations on `MotifHallu`.

do not follow the instructions, we compare the loss values by feeding the answers of "Yes" and "No", and take the one with a lower autoregressive language modeling loss as the answer. Following the practice in LVLMs, we present the F1 scores, accuracies, and the ratio that the model answers

| METHOD | BACE ↑ | BBBP ↑ | HIV ↑ | SIDER ↑ | ClinTox | MUV ↑ | Tox21 ↑ | CYP450 ↑ |
|---|---|---|---|---|---|---|---|---|
| # MOLECULES | 1,513 | 2,039 | 41,127 | 1,427 | 1,478 | 93,087 | 7,831 | 16,896 |
| # TASKS | 1 | 1 | 1 | 27 | 2 | 17 | 12 | 5 |
| *Specialist Models* | | | | | | | | |
| KV-PLM (Zeng et al., 2022) | 78.5 | 70.5 | 71.8 | 59.8 | 84.3 | 61.7 | 49.2 | 59.2 |
| GraphCL (You et al., 2020) | 75.3 | 69.7 | 78.5 | 60.5 | 76.0 | 69.8 | 73.9 | - |
| GraphMVP-C (Liu et al., 2022) | 81.2 | 72.4 | 77.0 | 60.6 | 84.5 | 74.4 | 77.1 | - |
| MoleculeSTM-G (Liu et al., 2023b) | 80.8 | 70.0 | 76.9 | 61.0 | 92.5 | 73.4 | 76.9 | - |
| MoMu (Su et al., 2022) | 76.7 | 70.5 | 75.9 | 60.5 | 79.9 | 60.5 | 57.8 | 58.0 |
| MolFM (Luo et al., 2023a) | 83.9 | **72.9** | 78.8 | 64.2 | 79.7 | 76.0 | 77.2 | - |
| Uni-Mol (Zhou et al., 2023) | **85.7** | **72.9** | **80.8** | 65.9 | 91.9 | 82.1 | 78.1 | - |
| Galactica-1.3B (Taylor et al., 2022) | 57.6 | 60.4 | 72.4 | 54.0 | 58.9 | 57.2 | 60.6 | 46.9 |
| Galactica-6.7B (Taylor et al., 2022) | 58.4 | 53.5 | 72.2 | 55.9 | 78.4 | - | 63.9 | - |
| Galactica-30B (Taylor et al., 2022) | 72.7 | 59.6 | **75.9** | 61.3 | 82.2 | - | 68.5 | - |
| Galactica-120B (Taylor et al., 2022) | 61.7 | 66.1 | 74.5 | 63.2 | 82.6 | - | 68.9 | - |
| GIMLET (Zhao et al., 2023) | 69.6 | 59.4 | 66.2 | - | - | 64.4 | 61.2 | 71.3 |
| *LLM Based Generalist Models* | | | | | | | | |
| LLama-2-7b-chat (4-shot) (Touvron et al., 2023b) | 76.9 | 54.2 | 67.8 | - | - | 46.9 | 62.0 | 57.6 |
| LLama-2-13b-chat (4-shot) (Touvron et al., 2023b) | 74.7 | 52.8 | **72.4** | - | - | 47.9 | 57.5 | 55.6 |
| InstructMol-G | 64.3 | 48.7 | 50.2 | 51.0 | 50.0 | 50.0 | 59.0 | 59.1 |
| **HIGHT-G** | **77.1** | **61.8** | 63.3 | **58.8** | **55.3** | **51.1** | **67.4** | **80.5** |

Table 2: ROC-AUC Results of molecular property prediction tasks (classification) on MoleculeNet (Wu et al., 2017). Evaluation on InstructMol and HIGHT adopt the likelihood of the tokens of "Yes" and "No". Most of the instruction tuning datasets are from GIMLET (Zhao et al., 2023). SIDER and ClinTox are converted following the MoleculeNet task description.

"Yes" (Li et al., 2023c). Given the severe imbalance of positive and negative samples in natural molecules, we separately report the F1 scores for positive and negative classes.

The results are given in Table 1, which show that the LGLMs with node-centric tokenization consistently answer with "Yes" despite the absence of the corresponding functional groups. In contrast, HIGHT significantly improves the worst class hallucinations up to 40% in terms of F1 scores, and the overall accuracies up to 30%, thereby reducing the hallucination of LGLMs to the functional groups that do not exist in the molecule.

We also conduct simple ablation studies by additionally incorporating the 1D sequence inputs with SELFIES following the literature (Fang et al., 2024; Cao et al., 2023). Contrary to previous results that additionally feeding the 1D sequence always improves the performance of LGLMs, We find that the additional 1D sequence may increase the degree of the hallucination. We suspect that it could be caused by the extremely long sequences of the SELFIES (Krenn et al., 2019) that may distract the attention signals of LLMs. Nevertheless, HIGHT suffers less from the distraction and performs better.

In addition, we also evaluate HIGHT without the tuning of HiPubChem. Aligned with our discussion in Sec. 3.2, HIGHT without HiPubChem will still suffer the hallucination, due to the low quality of the instruction tuning data. Interestingly, simply incorporating the hierarchical information at the architecture level can already help with the perception of graph information in LGLMs, which improves the robustness against hallucination, aligned with our discussion as in Sec. 4.1 (**?**).

## 5.3 MOLECULAR PROPERTY PREDICTION

In molecular property prediction, we leverage 8 datasets BACE, BBBP, HIV, SIDER, ClinTox, MUV, and Tox21 from MoleculeNet, and CYP450 from GIMLET (Zhao et al., 2023) to evaluate the classification performance with ROC-AUC. We also adopt the regression-based property prediction dataset from (Fang et al., 2024), where we evaluate several quantum chemistry measures such as HUMO, LUMO, and HUMO-LUMO gap (Ramakrishnan et al., 2014). The evaluation metric used to evaluate the regression based molecular property prediction is Mean Absolute Error (MAE). All the datasets are converted into instruction formats following previous works (Fang et al., 2024; Cao et al., 2023).

| METHOD | HOMO ↓ | LUMO ↓ | $\Delta\epsilon$ ↓ | AVG ↓ |
|---|---|---|---|---|
| *LLM Based Generalist Models* | | | | |
| Alpaca[†] (Dubois et al., 2023) | - | - | - | 322.109 |
| Baize[†] (Xu et al., 2023) | - | - | - | 261.343 |
| LLama2-7B (Touvron et al., 2023b) (5-shot ICL) | 0.7367 | 0.8641 | 0.5152 | 0.7510 |
| Vicuna-13B (Chiang et al., 2023) (5-shot ICL) | 0.7135 | 3.6807 | 1.5407 | 1.9783 |
| Mol-Instruction (Fang et al., 2024) | 0.0210 | 0.0210 | 0.0203 | 0.0210 |
| InstructMol-G | 0.0111 | 0.0133 | 0.0147 | 0.0130 |
| **HIGHT-G** | **0.0078** | **0.0086** | **0.0095** | **0.0086** |

Table 3: Results of molecular property prediction tasks (regression) on QM9. We report the result in MAE. †: few-shot in-context learning (ICL) results from (Fang et al., 2024). $\Delta\epsilon$ refers to the HOMO-LUMO energy gap.

| MODEL | BLEU-2↑ | BLEU-4↑ | ROUGE-1↑ | ROUGE-2↑ | ROUGE-L↑ | METEOR↑ |
|---|---|---|---|---|---|---|
| *Specialist Models* | | | | | | |
| MoT5-base (Edwards et al., 2022) | 0.540 | 0.457 | 0.634 | 0.485 | 0.568 | 0.569 |
| MoMu (MolT5-base) (Su et al., 2022) | 0.549 | 0.462 | - | - | - | 0.576 |
| MolFM (MolT5-base) (Luo et al., 2023a) | 0.585 | 0.498 | 0.653 | 0.508 | 0.594 | 0.607 |
| MolXPT (Liu et al., 2023c) | 0.594 | 0.505 | 0.660 | 0.511 | 0.597 | 0.626 |
| GIT-Mol-graph (Liu et al., 2024b) | 0.290 | 0.210 | 0.540 | 0.445 | 0.512 | 0.491 |
| GIT-Mol-SMILES (Liu et al., 2024b) | 0.264 | 0.176 | 0.477 | 0.374 | 0.451 | 0.430 |
| GIT-Mol-(graph+SMILES) (Liu et al., 2024b) | 0.352 | 0.263 | 0.575 | 0.485 | 0.560 | 0.430 |
| Text+Chem T5-augm-base (Christofidellis et al., 2023) | **0.625** | **0.542** | **0.682** | **0.543** | **0.622** | **0.648** |
| *Retrieval Based LLMs* | | | | | | |
| GPT-3.5-turbo (10-shot MolReGPT) (Li et al., 2023a) | 0.565 | 0.482 | 0.623 | 0.450 | 0.543 | 0.585 |
| GPT-4-0314 (10-shot MolReGPT) (Li et al., 2023a) | 0.607 | 0.525 | 0.634 | 0.476 | 0.562 | 0.610 |
| *LLM Based Generalist Models* | | | | | | |
| GPT-3.5-turbo (zero-shot) (Li et al., 2023a) | 0.103 | 0.050 | 0.261 | 0.088 | 0.204 | 0.161 |
| BioMedGPT-10B (Luo et al., 2023b) | 0.234 | 0.141 | 0.386 | 0.206 | 0.332 | 0.308 |
| Mol-Instruction (Fang et al., 2024) | 0.249 | 0.171 | 0.331 | 0.203 | 0.289 | 0.271 |
| InstructMol-G | 0.481 | 0.381 | 0.554 | 0.379 | 0.488 | 0.503 |
| **HIGHT-G** | **0.504** | **0.405** | **0.570** | **0.397** | **0.502** | **0.524** |

Table 4: Results of molecular description generation task on the test split of ChEBI-20.

The results of molecular property prediction are given in Table 2 and Table 3 for classification and regression, respectively. We can find that, no matter for classification or regression-based molecular property prediction, HIGHT always significantly boosts the performance. The improvements brought by HIGHT serve as strong evidence verifying our discussion in Sec. 4 about the importance of hierarchical information for graph-language alignment. Remarkably, in CYP450 (Zhao et al., 2023), HIGHT significantly outperforms the state-of-the-art specialist model, demonstrating the advances of LGLM with hierarchical graph tokenization. Interestingly, Llama-2 (Touvron et al., 2023b) can match the state-of-the-art performance in HIV in a few-shot setting, while performing significantly worse in other datasets, for which we suspect there might exist some data contamination.

## 5.4 MOLECULAR DESCRIPTION GENERATION

For the task of molecular description generation or molecular captioning, we adopt the widely used benchmark ChEBI-20 (Edwards et al., 2021). Given the molecules, ChEBI-20 evaluates the linguistic distances of the generated molecule captions of molecular characteristics such as structure, properties, biological activities etc.. Following the common practice, we report the metrics of BLEU (Papineni et al., 2002), ROUGE (Lin, 2004) and Meteor (Banerjee & Lavie, 2005). The LGLMs are trained using the ChEBI-20 train split and evaluated using the test split. The final is selected according to the best training loss.

The results are given in Table 4. We can find that HIGHT consistently brings significant improvements over LGLMs with node-centric tokenization. Nevertheless, compared to the specialist models such as MoT5 (Edwards et al., 2022) that are pretrained on a significant amount of molecule-text related corpus, there remains a gap for generalist LGLMs even with HIGHT. The gap calls for interesting future investigations on how to incorporate HIGHT into the pretraining of the LGLMs properly.

## 5.5 CHEMICAL REACTION PREDICTION

For chemical reaction prediction tasks, we incorporate three tasks from Mol-Instructions (Fang et al., 2024), i.e., reagent prediction, forward reaction prediction, and retrosynthesis prediction, which are crucial for AI-aided drug discovery. Reagent prediction aims to predict the suitable reagents for a particular chemical reaction. Forward reaction prediction aims to predict the products of a chemical reaction, given the reactants and the reagents. Retrosynthesis prediction aims to predict the suitable reactants given a target product. The inputs and outputs for chemical reaction related tasks adopt the SELFIES (Krenn et al., 2019) as recommended by (Fang et al., 2024). In terms of the evaluation metrics, we incorporate both linguistic distance metrics such as BLEU (Papineni et al., 2002) and Levenshtein (Yujian & Bo, 2007), as well as molecular similarity measures such as similarity of the molecular fingerprints by RDKit (Landrum, 2016).

The results are given in Table 5. It can be found that across all tasks in chemical reaction prediction, LGLMs with HIGHT consistently and significantly improve the performances compared to the node-centric tokenization. Meanwhile, LGLMs with HIGHT achieve state-of-the-art results in several tasks and metrics, compared to other generalist models that even incorporate a stronger LLM backbone such as Mol-Instruction, and additional information of SELFIES.

| MODEL | EXACT↑ | BLEU↑ | LEVENSHTEIN↓ | RDK FTS↑ | MACCS FTS↑ | MORGAN FTS↑ | VALIDITY↑ |
|---|---|---|---|---|---|---|---|
| *Reagent Prediction* | | | | | | | |
| Alpaca† (Dubois et al., 2023) | 0.000 | 0.026 | 29.037 | 0.029 | 0.016 | 0.001 | 0.186 |
| Baize† (Xu et al., 2023) | 0.000 | 0.051 | 30.628 | 0.022 | 0.018 | 0.004 | 0.099 |
| ChatGLM† (Zeng et al., 2023) | 0.000 | 0.019 | 29.169 | 0.017 | 0.006 | 0.002 | 0.074 |
| LLama† (Touvron et al., 2023a) | 0.000 | 0.003 | 28.040 | 0.037 | 0.001 | 0.001 | 0.001 |
| Vicuna† (Chiang et al., 2023) | 0.000 | 0.010 | 27.948 | 0.038 | 0.002 | 0.001 | 0.007 |
| Mol-Instruction (Fang et al., 2024) | 0.044 | 0.224 | **23.167** | 0.237 | **0.364** | 0.213 | 1.000 |
| LLama-7b* (Touvron et al., 2023a)(LoRA) | 0.000 | 0.283 | 53.510 | 0.136 | 0.294 | 0.106 | 1.000 |
| InstructMol-G | 0.031 | 0.429 | 31.447 | 0.389 | 0.249 | 0.220 | 1.000 |
| **HIGHT-G** | 0.050 | **0.462** | 28.970 | **0.441** | 0.314 | **0.275** | 1.000 |
| *Forward Reaction Prediction* | | | | | | | |
| Alpaca† (Dubois et al., 2023) | 0.000 | 0.065 | 41.989 | 0.004 | 0.024 | 0.008 | 0.138 |
| Baize† (Xu et al., 2023) | 0.000 | 0.044 | 41.500 | 0.004 | 0.025 | 0.009 | 0.097 |
| ChatGLM† (Zeng et al., 2023) | 0.000 | 0.183 | 40.008 | 0.050 | 0.100 | 0.044 | 0.108 |
| LLama† (Touvron et al., 2023a) | 0.000 | 0.020 | 42.002 | 0.001 | 0.002 | 0.001 | 0.039 |
| Vicuna† (Chiang et al., 2023) | 0.000 | 0.057 | 41.690 | 0.007 | 0.016 | 0.006 | 0.059 |
| Mol-Instruction (Fang et al., 2024) | 0.045 | 0.654 | 27.262 | 0.313 | 0.509 | 0.262 | 1.000 |
| LLama-7b* (Touvron et al., 2023a)(LoRA) | 0.012 | 0.804 | 29.947 | 0.499 | **0.649** | **0.407** | 1.000 |
| InstructMol-G | 0.031 | 0.853 | 24.790 | 0.512 | 0.362 | 0.303 | 0.993 |
| **HIGHT-G** | 0.037 | **0.869** | **23.759** | **0.590** | 0.394 | 0.340 | 0.993 |
| *Retrosynthesis* | | | | | | | |
| Alpaca† (Dubois et al., 2023) | 0.000 | 0.063 | 46.915 | 0.005 | 0.023 | 0.007 | 0.160 |
| Baize† (Xu et al., 2023) | 0.000 | 0.095 | 44.714 | 0.025 | 0.050 | 0.023 | 0.112 |
| ChatGLM† (Zeng et al., 2023) | 0.000 | 0.117 | 48.365 | 0.056 | 0.075 | 0.043 | 0.046 |
| LLama† (Touvron et al., 2023a) | 0.000 | 0.036 | 46.844 | 0.018 | 0.029 | 0.017 | 0.010 |
| Vicuna† (Chiang et al., 2023) | 0.000 | 0.057 | 46.877 | 0.025 | 0.030 | 0.021 | 0.017 |
| Mol-Instruction (Fang et al., 2024) | 0.009 | 0.705 | 31.227 | 0.283 | 0.487 | 0.230 | 1.000 |
| LLama-7b* (Touvron et al., 2023a)(LoRA) | 0.000 | 0.283 | 53.510 | 0.136 | 0.294 | 0.106 | 1.000 |
| InstructMol-G | 0.001 | 0.835 | 31.359 | 0.447 | 0.277 | 0.241 | 0.996 |
| **HIGHT-G** | 0.008 | **0.863** | 28.912 | **0.564** | **0.340** | **0.309** | 1.000 |

Table 5: Results of chemical reaction tasks. These tasks encompass reagent prediction, forward reaction prediction, and retrosynthesis. †: few-shot ICL results from Fang et al. (2024). ∗: use task-specific instruction data to finetune.

## 5.6 ABLATION STUDIES

To better understand the effectiveness of distinct components in HIGHT, we conduct additional ablation studies that train InstructMol (Cao et al., 2023) with HiPubChem, or with the laplacian positional encodings in molecular captioning tasks. The results are given in Table 6. We can find that, merely incorporating positional encoding or hierarchical instruction tuning is not sufficient to achieve the same performance as HIGHT. On the contrary, without a proper architecture design as HIGHT, LGLMs with previous node-centric tokenization with HiPubChem will confuse LLMs and even lead to degenerated downstream task performances.

| METHODS | BLEU-2↑ | BLEU-4↑ | ROUGE-1↑ | ROUGE-2↑ | ROUGE-L↑ | METEOR↑ |
|---|---|---|---|---|---|---|
| InstructMol+G | 0.481 | 0.381 | 0.554 | 0.379 | 0.488 | 0.503 |
| +Positional Encoding | 0.488 | 0.388 | 0.556 | 0.383 | 0.491 | 0.508 |
| +HiPubChem | 0.473 | 0.372 | 0.547 | 0.371 | 0.481 | 0.495 |
| **HIGHT-G** | **0.504** | **0.405** | **0.570** | **0.397** | **0.502** | **0.524** |

Table 6: Abaltion study in Molecule Description Generation task.

## 6 CONCLUSIONS

This paper presents HIGHT, a novel hierarchical graph tokenization technique, which enhances the synergy between molecules and language. By incorporating the hierarchical graph information, HIGHT improves the graph-language alignment performance, reducing hallucinations and boosting accuracy in molecular tasks, as validated through comprehensive benchmarking. Nevertheless, the current concentration on molecular graphs requires further verification for wider applicability to other forms of graph data, such as that originated from social networks. Despite the limitation, HIGHT represents a significant step forward in advancing LLMs' graph comprehension capability, and highlighting paths for future research in this direction.

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

# Appendix of HIGHT

CONTENTS

## A   BROADER IMPACTS

This paper mainly focuses on how to best represent graph information for LLMs to understand better about the graphs. We demonstrate the effectiveness of our method on molecule-centric tasks which could facilitate the broader use of LLMs for tasks like AI-aided drug discovery and human-machine interactions in biomedicine. Besides, this paper does not raise any ethical concerns. This study does not involve any human subjects, practices to data set releases, potentially harmful insights, methodologies and applications, potential conflicts of interest and sponsorship, discrimination/bias/fairness concerns, privacy and security issues, legal compliance, and research integrity issues.

## B   DETAILS OF INSTRUCTION TUNING DATASETS

We provide a summary of the datasets for instruction tuning and evaluation in this paper as in Table 7. Meanwhile, we also list the data sources and the corresponding licenses of the sources for each task and dataset. Then, we will elaborate more on the details of the datasets in the following subsections.

Table 7: Summary of datasets involved in our paper.

| Datasets | Train | Test | Content |
|---|---|---|---|
| PubChem | 295,228 | N/A | Molecules and the associated descriptions from PubChem. |
| HiPubChem | 295,228 | N/A | Molecules and the associated descriptions from PubChem and about functional groups in the molecule. |
| MoleculeNet-HIV | 32,901 | 4,113 | Question answering about the ability of the molecule to inhibit HIV replication. |
| MoleculeNet-BACE | 1,210 | 152 | Question answering about the ability of the molecule to bind to the BACE1 protein |
| MoleculeNet-BBBP | 1,631 | 204 | Question answering about the ability of the molecule to diffuse across the brain blood barrier. |
| MoleculeNet-SIDER | 1,141 | 143 | Question answering about the ability of the side effects. |
| MoleculeNet-ClinTox | 1,188 | 148 | Question answering about the toxicology. |
| MoleculeNet-MUV | 74,469 | 9,309 | Question answering about PubChem bioAssay |
| MoleculeNet-Tox21 | 6,877 | 860 | Question answering about Toxicology in the 21st century |
| CYP45- | 13,516 | 1,690 | Question answering about CYP PubChem BioAssay CYP 1A2, 2C9, 2C19, 2D6, 3A4 inhibition. |
| Property Prediction (Regression) | 360,113 | 1,987 | Question answering about the quantum mechanics properties of the molecule. |
| Forward Reaction Prediction | 124,384 | 1,000 | Question answering about the products of a chemical reaction, given specific reactants and reagents. |
| Reagent Prediction | 124,384 | 1,000 | Question answering about suitable catalysts, solvents, or ancillary substances required for a specific chemical reaction. |
| Retrosynthesis Prediction | 128,684 | 1,000 | Question answering about the reactants and reagents of a chemical reaction, given specific products. |
| ChEBI-20 | 26,407 | 3,300 | Molecules and the associated Chemical Entities of Biological Interest (ChEBI) (Hastings et al., 2015) annotations. |
| MotifHallu | N/A | 23,924 | Question answering about existing functional groups in the molecule. |

### B.1   DETAILS OF THE PUBCHEM DATASET

PubChem[2] is one of the largest public molecule database (Kim et al., 2022), and has been widely adopted by the alignment training of LGLMs (Liu et al., 2023d;b; Cao et al., 2023). Our construction of PubChem predominantly follows Liu et al. (2023b). We will briefly describe the main steps and interested readers may refer the details to (Liu et al., 2023b):

---

[2]https://pubchem.ncbi.nlm.nih.gov

Table 8: Summary of data resources and licenses of datasets involved in our paper.

| Tasks/Datasets | Data Sources | License URL | License Note |
|---|---|---|---|
| PubChem, HiPubChem | PubChem | https://www.nlm.nih.gov/web_policies.html | Works produced by the U.S. government are not subject to copyright protection in the United States. Any such works found on National Library of Medicine (NLM) Web sites may be freely used or reproduced without permission in the U.S. |
| Reaction Prediction | USPTO | https://www.uspto.gov/learning-and-resources/open-data-and-mobility | It can be freely used, reused, and redistributed by anyone. |
| Property Prediction | MoleculeNet | https://opensource.org/license/mit/ | Permission is hereby granted, free of charge, to any person obtaining a copy of this software and associated documentation files (the "Software"), to deal in the Software without restriction, including without limitation the rights to use, copy, modify, merge, publish, distribute, sublicense, and/or sell copies of the Software, and to permit persons to whom the Software is furnished to do so. |
| Property Prediction | CYP450 | https://www.nlm.nih.gov/web_policies.html | The data is from Zhao et al. (2023) that curates PubChem BioAssay CYP 1A2, 2C9, 2C19, 2D6, 3A4 inhibition. Thus it shares the same license as PubChem. |
| Molecular Description, MotifHallu | ChEBI | https://creativecommons.org/licenses/by/4.0/ | You are free to: Share — copy and redistribute the material in any medium or format. Adapt — remix, transform, and build upon the material for any purpose, even commercially. |

- We curate the data from PubChem using the official API and set the data cutoff date as 12 Jan. 2024. It downloads both the molecular structure (e.g., SMILES, 2D molecular graphs) in SDF format, and the text descriptions.
- Then, we will filter out molecules that do not have descriptions or can not match via the PubChem ID. In the descriptions, the molecule names are replaced with "This molecule", in order to facilitate LLMs to understand the instructions.

Finally, the curation generates 295k molecule-text pairs that we term as PubChem-295k. PubChem-295k will be mainly used for the stage 1 alignment training.

Table 9: Examples of PubChem and HiPubChem datasets.

| PubChem | HiPubChem |
|---|---|
| *SMILES: CC(=O)OC(CC(=O)[O-])C[N+](C)(C)C* | |
| This molecule is an O-acylcarnitine having acetyl as the acyl substituent. It has a role as a human metabolite. It is functionally related to an acetic acid. It is a conjugate base of an O-acetylcarnitinium. | This molecule has 1 carboxylic acids functional group. This molecule has no methyl amide, or amide, or nitro or thiols groups. This molecule is an O-acylcarnitine having acetyl as the acyl substituent. It has a role as a human metabolite. It is functionally related to an acetic acid. It is a conjugate base of an O-acetylcarnitinium. |
| *SMILES: CCN(CC)CCOC(=O)C(Cc1cccc2ccccc12)CC1CCCO1* | |
| This molecule is a member of naphthalenes. | This molecule has 0 functional groups. This molecule is a member of naphthalenes. |
| *SMILES: Cc1c2[nH]c(c1CCC(=O)O)Cc1[nH]c(c(CCC(=O)O)c1C)Cc1[nH]c(c(CCC(=O)O)c1C)Cc1[nH]c(c(C)c1CCC(=O)O)C2* | |
| This molecule is a coproporphyrinogen. It has a role as an Escherichia coli metabolite and a mouse metabolite. It is a conjugate acid of a coproporphyrinogen III(4-). | This molecule has 1 carboxylic acids functional groups. This molecule has no methyl amide, or diazo, or cyano or thiols groups. This molecule is a coproporphyrinogen. It has a role as an Escherichia coli metabolite and a mouse metabolite. It is a conjugate acid of a coproporphyrinogen III(4-). |

## B.2 DETAILS OF HIPUBCHEM DATASET

HiPubChem augments the molecular instruction tuning dataset with captions of the functional groups. We consider both the positive and negative appearances of motifs when augmenting the instructions. For the positive case, we directly append the caption of all functional groups detected with RDKit:

```
        This molecule has <#> of <functional group name> groups.
```

For the negative case, we randomly sample $k_{neg}$ that do not appear in the molecule:

```
        This molecule has no <functional group name> groups.
```

Despite the simple augmentation strategy, we find that `HiPubChem` significantly reduces the hallucination issue, and improves the molecule-language alignment performance.

For comparison, we provide examples of PubChem and `HiPubChem` in Table 9.

### B.3 DETAILS OF PROPERTY PREDICTION DATASET

The task of molecular property prediction mainly aims to predict certain biochemical or physical properties of molecules. Usually, these properties have a close relation with the molecular substructures (i.e., functional groups) (Bohacek et al., 1996). In this work, we consider the scenarios of both binary classification based and the regression based molecular property prediction, and the datasets are mainly derived from MoleculeNet (Wu et al., 2017).

For the classification, we consider three subtasks, HIV, BACE, and BBBP. The HIV subtask mainly evaluates whether the molecule is able to impede the replication of the HIV virus. The BACE subtask mainly evaluates the binding capability of a molecule to the BACE1 protein. The BBBP subtask mainly evaluates the capability of a molecule to passively diffuse across the human brain blood barrier. For task-specific instruction tuning, we convert those classification based datasets into instructions. Examples are given in Table 10.

Table 10: Examples of the property prediction (classification) datasets.

| Dataset | Question | Answer |
|---------|----------|--------|
| HIV | *SMILES: N=C1OC2(c3ccccc3)C3=C(OC(=NC)N2C)C(=O)OC3(c2ccccc2)N1C* | |
| | Please help me evaluate whether the given molecule can impede the replication of the HIV virus. | No |
| BACE | *SMILES: CN(C(=O)CCc1cc2ccccc2nc1N)C1CCCCC1* | |
| | Can the given molecule bind to the BACE1 protein? | Yes |
| BBBP | *SMILES: Cc1c[nH+][o+]c(C([NH])CC(C)C(C)N(C(C)(C)C)C(C)(N)N)c1[O-]* | |
| | Can the given molecule passively diffuse across the brain blood barrier? | Yes |

Table 11: Examples of the property prediction (regression) datasets.

| Question | Answer |
|----------|--------|
| *SELFIES: [O][=C][O][C][C][C][C][Ring1][=Branch1][C][Ring1][Ring2]* | |
| Can you give me the energy difference between the HOMO and LUMO orbitals of this molecule? | 0.2756 |
| *SELFIES: [C][C][C][=Branch1][C][=O][N][Branch1][C][C][C][=Branch1][C][=O][N]* | |
| What is the lowest unoccupied molecular orbital (LUMO) energy of this molecule? | -0.0064 |
| *SELFIES: [C][C][=C][O][C][=C][Ring1][Branch1][C][Branch1][C][C][C]* | |
| Please provide the highest occupied molecular orbital (HOMO) energy of this molecule. | -0.2132 |

For regression, we adopt the instruction tuning data from `Mol-Instructions` (Fang et al., 2024). The regression based property prediction focuses on predicting the quantum mechanics properties of the molecules. The 1D sequence information in this task is given by SELFIES (Krenn et al., 2019). The original data is sourced from the QM9 subset of the MolculeNet (Wu et al., 2017). There are three subtasks: (i) Highest occupied molecular orbital (HOMO) energy prediction; (ii) Lowest occupied molecular orbital (LUMO) energy prediction; (iii) and HUMO-LUMO gap energy prediction. Some examples of the regression based property prediction dataset are given in Table 11.

### B.4 DETAILS OF REACTION PREDICTION DATASET

We adopt three chemical reaction related tasks from `Mol-Instructions` (Fang et al., 2024): Forward reaction prediction, reagent prediction, and retrosynthesis prediction. The input and output contain 1D sequence information given by SELFIES (Krenn et al., 2019). Some examples of the

`Mol-Instructions` datasets are given in Table 12, where the SELFIES represented molecules are denoted as "<SELFIES>" for clarity.

Table 12: Examples of the chemical reaction datasets.

| Task | Examples |
|---|---|
| Forward Reaction Prediction | *Question:* With the provided reactants and reagents, propose a potential product.*<SELFIES>* |
| | *Answer: <SELFIES>* |
| Reagent Prediction | *Question:* Please suggest some possible reagents that could have been used in the following chemical reaction. The reaction is *<SELFIES>* |
| | *Answer: <SELFIES>* |
| Retrosynthesis Prediction | *Question:* Please suggest potential reactants for the given product. The product is: *<SELFIES>* |
| | *Answer: <SELFIES>* |

The task of forward reaction prediction aims to predict the possible products of a chemical reaction. The input includes the SELFIES sequences of the reactant and reagent of the chemical reaction. And the model needs to predict the SELFIES of the products. The original data is sourced from USPTO [3], which consists of chemical reactions of organic molecules extracted from American patents and patent applications.

The task of reagent reaction prediction aims to predict the suitable catalysts, solvents, and ancillary substances with respect to a chemical reaction. The input includes the SELFIES sequences of the chemical reaction. The original data is sourced from USPTO [4], as the other tasks.

The task of retrosynthesis prediction aims to reverse engineer a particular compound by predicting the potential reactants or reagents that are required to synthesis the compound. The input includes the SELFIES sequences of the target product. The original data is sourced from USPTO [5], similar to the other tasks.

## B.5 DETAILS OF MOLECULAR DESCRIPTION DATASET

For the molecular description task, we adopt a widely used dataset `ChEBI-20` (Edwards et al., 2021). Based on the molecules from PubChem, Edwards et al. (2021) collected the Chemical Entities of Biological Interest (ChEBI) (Hastings et al., 2015) annotations of the molecules, which are the descriptions of molecules. We transform the task into the instructions, and present some samples in Table 13. The authors collect $33,010$ molecule-text pairs and split them into training ($80\%$), validation ($10\%$), and testing ($10\%$) subsets. We mainly adopt the original training split to tune the model and evaluate the tuned model on the original test split.

Table 13: Examples of the molecular description datasets.

| Question | Answer |
|---|---|
| *SMILES: C1=CC=C(C=C1)[As](=O)(O)[O-]* Could you give me a brief overview of this molecule? | The molecule is the organoarsonic acid anion formed by loss of a single proton from the arsonic acid grouping in phenylarsonic acid. It is a conjugate base of a phenylarsonic acid. |
| *SMILES: CCCCCCCCCCCC(=O)OC(=O)CCCCCCCCCCC* Could you provide a description of this molecule? | The molecule is an acyclic carboxylic anhydride resulting from the formal condensation of the carboxy groups of two molecules of dodecanoic acid. It derives from a dodecanoic acid. |
| *SMILES: CCCCNC=O* Please give me some details about this molecule. | The molecule is a member of the class of formamides that is formamide substituted by a butyl group at the N atom. It has a role as a human metabolite. It derives from a formamide. |

---

[3]`https://developer.uspto.gov/data`
[4]`https://developer.uspto.gov/data`
[5]`https://developer.uspto.gov/data`

### B.6  DETAILS OF MOTIFHALLU DATASET

The `MotifHallu` is mainly used to measure the hallucination of common functional groups by LGLMs. For the construction of `MotifHallu`, we consider the common functional groups in RDKit[6] as shown in Table 14. There are 39 common functional groups, while we neglect the one with the name of "???".

Then, we leverage RDKit (Landrum, 2016) to detect the existence of the left 38 valid functional groups within a molecule. We consider $3,300$ molecules from `ChEBI-20` test split (Edwards et al., 2021), and adopt the query style as for large vision-language models (Li et al., 2023c) that queries the existence of specific functional group one by one:

> Is there a <functional group name> in the molecule?

Examples of `MotifHallu` are given in Table 15.

During the evaluation, we detect whether the LGLM gives outputs meaning "Yes" or "No" following the practice in (Li et al., 2023c). For each molecule, we construct questions with positive answers for all kinds of functional groups detected in the molecule, and questions with negative answers for randomly sampled 6 functional groups from the 38 common functional groups in RDKit. The construction finally yields $23,924$ query answer pairs about the existence of functional groups in the molecule. While it is easy to scale up `MotifHallu` by automatically considering more molecules and a broader scope of functional groups, we find that the current scale is already sufficient to demonstrate the hallucination phenomena in LGLMs.

## C  DETAILS OF EXPERIMENTS

**Implementation of graph tokenizer.**  We implement the GNN tokenizer/encoder based on the same GNN backbone, which is a 5-layer GIN (Xu et al., 2019). The hidden dimension is 300. For the node-centric tokenization, we employ the VQVAE GNN tokenizer from `Mole-BERT` (Xia et al., 2023) and adopt self-supervised learning tasks from the official `Mole-BERT` implementation.[7] For `HIGHT`, we train the VQVAE with the self-supervised learning tasks from (Zang et al., 2023) based on the official implementation.[8] Meanwhile, we set the hyperparameters of GNN tokenizer training the same as those recommended by (Xia et al., 2023; Zang et al., 2023).

After training the tokenizer, we adopt the GNN encoder within the tokenizer instead of the codebook embeddings as we empirically find that the GNN embeddings perform better than that using the VQVAE codebook embeddings.

**Implementation of LGLMs.**  For the cross-modal adapters, we implement it as a single-layer MLP with an input dimension of 300 as our main focus is the tokenization. For `HIGHT`, we adopt three distinct adapters to handle the node-level, motif-level and graph-level embeddings. Meanwhile, we also adopt a Laplacian position encodings with respect to the supernode-augmented graphs. The dimension of the Laplacian position encoding is set to 8, therefore the input dimensions of the adapters in `HIGHT` will be 308.

For the LoRA adapters, we use a LoRA rank of 128 and a scaling value $\alpha$ of 256 (Hu et al., 2022) for all methods and tasks.

For the base LLM, we mainly adopt `vicuna-v-1.3-7B` (Chiang et al., 2023). The overall scale of parameters is around 6.9B.

**Implementation of instruction tuning.**  In stage 1 instruction tuning, we train all methods based on PubChem-295k dataset. The training goes 5 epochs, with a batch size of 64 (distributed to 4 GPUs) by default. If there is an OOM issue, we will decrease the batch size a little bit to 40. The learning rate is set to $2 \times 10^{-3}$ for all methods.

---

[6] https://github.com/rdkit/rdkit/blob/master/Data/FunctionalGroups.txt
[7] https://github.com/junxia97/Mole-BERT
[8] https://github.com/ZangXuan/HiMol

Table 14: List of functional groups from RDKit used to construct `MotifHallu`. The functional group with the name "???" is neglected.

| Chemical Representation | SMARTS | Name |
|---|---|---|
| -NC(=O)CH3 | *-[N;D2]-[C;D3](=O)-[C;D1;H3] | methyl amide |
| -C(=O)O | *-C(=O)[O;D1] | carboxylic acids |
| -C(=O)OMe | *-C(=O)[O;D2]-[C;D1;H3] | carbonyl methyl ester |
| -C(=O)H | *-C(=O)-[C;D1] | terminal aldehyde |
| -C(=O)N | *-C(=O)-[N;D1] | amide |
| -C(=O)CH3 | *-C(=O)-[C;D1;H3] | carbonyl methyl |
| -N=C=O | *-[N;D2]=[C;D2]=[O;D1] | isocyanate |
| -N=C=S | *-[N;D2]=[C;D2]=[S;D1] | isothiocyanate |
| *Nitrogen containing groups* | | |
| -NO2 | *-[N;D3](=[O;D1])[O;D1] | nitro |
| -N=O | *-[N;R0]=[O;D1] | nitroso |
| =N-O | *=[N;R0]-[O;D1] | oximes |
| =NCH3 | *=[N;R0]-[C;D1;H3] | Imines |
| -N=CH2 | *-[N;R0]=[C;D1;H2] | Imines |
| -N=NCH3 | *-[N;D2]=[N;D2]-[C;D1;H3] | terminal azo |
| -N=N | *-[N;D2]=[N;D1] | hydrazines |
| -N#N | *-[N;D2]#[N;D1] | diazo |
| -C#N | *-[C;D2]#[N;D1] | cyano |
| *S containing groups* | | |
| -SO2NH2 | *-[S;D4](=[O;D1])(=[O;D1])-[N;D1] | primary sulfonamide |
| -NHSO2CH3 | *-[N;D2]-[S;D4](=[O;D1])(=[O;D1])-[C;D1;H3] | methyl sulfonamide |
| -SO3H | *-[S;D4](=O)(=O)-[O;D1] | sulfonic acid |
| -SO3CH3 | *-[S;D4](=O)(=O)-[O;D2]-[C;D1;H3] | methyl ester sulfonyl |
| -SO2CH3 | *-[S;D4](=O)(=O)-[C;D1;H3] | methyl sulfonyl |
| -SO2Cl | *-[S;D4](=O)(=O)-[Cl] | sulfonyl chloride |
| -SOCH3 | *-[S;D3](=O)-[C;D1] | methyl sulfinyl |
| -SCH3 | *-[S;D2]-[C;D1;H3] | methylthio |
| -S | *-[S;D1] | thiols |
| =S | *=[S;D1] | thiocarbonyls |
| *Miscellaneous fragments* | | |
| -X | *-[#9,#17,#35,#53] | halogens |
| -tBu | *-[C;D4]([C;D1])([C;D1])-[C;D1] | t-butyl |
| -CF3 | *-[C;D4](F)(F)F | trifluoromethyl |
| -C#CH | *-[C;D2]#[C;D1;H] | acetylenes |
| -cPropyl | *-[C;D3]1-[C;D2]-[C;D2]1 | cyclopropyl |
| *Teeny groups* | | |
| -OEt | *-[O;D2]-[C;D2]-[C;D1;H3] | ethoxy |
| -OMe | *-[O;D2]-[C;D1;H3] | methoxy |
| -O | *-[O;D1] | side-chain hydroxyls |
| =O | *=[O;D1] | side-chain aldehydes or ketones |
| -N | *-[N;D1] | primary amines |
| =N | *=[N;D1] | ??? |
| #N | *#[N;D1] | nitriles |

For classification-based property prediction, the training goes 20 epochs, with a batch size of 128 (distributed to 4 GPUs) by default. If there is an OOM issue, we will decrease the batch size a little bit to 64. The learning rate is set to $8 \times 10^{-5}$ for all methods.

For regression-based property prediction, the training goes 5 epochs, with a batch size of 64 (distributed to 4 GPUs) by default. The learning rate is set to $2 \times 10^{-5}$ for all methods.

Table 15: Examples of the `MotifHallu` dataset.

| Question | Answer |
|---|---|
| *SMILES: COC1=CC=CC2=C1C(=CN2)C/C(=N/OS(=O)(=O)[O-])/S[C@H]3[C@@H]([C@H]([C@@H]([C@H](O3)CO)O)O)O* | |
| Is there a methyl ester sulfonyl group in the molecule? | No |
| *SMILES: CN(C)C(=O)C(CCN1CCC(CC1)(C2=CC=C(C=C2)Cl)O)(C3=CC=CC=C3)C4=CC=CC=C4* | |
| Is there a carbonyl methyl ester group in the molecule? | Yes |
| *SMILES: CN(C)C(=O)C(CCN1CCC(CC1)(C2=CC=C(C=C2)Cl)O)(C3=CC=CC=C3)C4=CC=CC=C4* | |
| Is there a terminal aldehyde group in the molecule? | No |

For molecular description, the training goes 50 epochs, with a batch size of 64 (distributed to 4 GPUs) by default. If there is an OOM issue, we will decrease the batch size a little bit to 32. The learning rate is set to $8 \times 10^{-5}$ for all methods.

For forward reaction prediction, the training goes 5 epochs, with a batch size of 64 (distributed to 4 GPUs) by default. The learning rate is set to $2 \times 10^{-5}$ for all methods.

For reagent prediction, the training goes 5 epochs, with a batch size of 64 (distributed to 4 GPUs) by default. The learning rate is set to $2 \times 10^{-5}$ for all methods.

For retrosynthesis prediction, the training goes 5 epochs, with a batch size of 64 (distributed to 4 GPUs) by default. The learning rate is set to $2 \times 10^{-5}$ for all methods.

**Training and evaluation.** Throughout the paper, we use a max token length of 2048. Meanwhile, we adopt an AdamW optimizer with a warmup ratio of 3% for optimizing all models. We select the final model according to the best training loss.

For the evaluation of classification-based property prediction, we adopt the ROC-AUC following the common practice (Wu et al., 2017).

For the evaluation of regression-based property prediction, we adopt the Mean Absolute Error (MAE) following the common practice (Fang et al., 2024).

For the evaluation of molecular description, we adopt BLEU-2, BLEU-4, ROUGE-1, ROUGE-2, ROUGE-L, and METEOR following the common practice (Papineni et al., 2002; Lin, 2004; Edwards et al., 2021). To improve the reliability of the evaluation, the metrics are computed based on the tokenizer `scibert_scivocab_uncased` of SciBERT (Beltagy et al., 2019).

We follow the common practice to evaluate models for the tasks of chemical reaction predictions (Fang et al., 2024). We adopt linguistic metrics such as BLEU (Papineni et al., 2002), ROUGE-L (Lin, 2004), METEOR (Banerjee & Lavie, 2005) and Levenshtein scores (Yujian & Bo, 2007). Meanwhile, we also validate the validity of the generated molecular sequences with RDKit (Landrum, 2016). In addition, several molecular similarity measures are also leveraged. Specifically, we present the MAE of the RDKit, MACCS, and Morgan fingerprints to assess the semantic similarity of the generated compounds and the ground truth ones (Durant et al., 2002; Schneider et al., 2015).

As for the `MotifHallu`, in order to avoid the drawbacks that LGLMs may output answers that do not follow the instructions, we compare the loss values by feeding the answers of "Yes" and "No", and take the one with a lower autoregressive language modeling loss as the answer. Following the practice in LVLMs, we present the F1 scores, accuracies, and the ratio that the model answers "Yes" (Li et al., 2023c). Given the severe imbalance of positive and negative samples, we separately report the F1 scores for positive and negative classes.

**Software and hardware.** We implement our methods with PyTorch 11.3 (Paszke et al., 2019). We run experiments on Linux Servers with NVIDIA V100 and NVIDIA A100 (40G) graphics cards with CUDA 11.7.

