# OpenReview forum: "Improving Molecule-Language Alignment with Hierarchical Graph Tokenization"
_ICLR.cc/2025/Conference — Submitted to ICLR 2025_

### Official Review · Reviewer_mY8L · 2024-10-28

**Soundness:** 2
**Presentation:** 3
**Contribution:** 2
**Rating:** 3
**Confidence:** 3

**Summary:**

This paper proposes a framework named HIGHT to align molecular data with LLMs. It identifies a shortcoming of LLM on learning functional groups, and proposes to extend the graph tokenization to motif level. Specifically, its input to the LLM includes node/atom embeddings as well as motif embeddings. The model is fine-tuned with motif prediction tasks on a dataset constructed using RDKit. The model shows good performance on molecule properties prediction compared to language models.

**Strengths:**

The overall presentation is clear and the paper is easy to follow. The work proposes a complete pipeline to build a model with stronger motif/functional group querying ability. Using motif tokens is a straight-forward solution to enhance such ability. Various experiments are conducted to validate the model.

**Weaknesses:**

- From the novelty and contribution perspective, taking motif representations/tokens is not new. By simply searching on Google, I found several papers that extract motifs for graph modeling [1, 2] (as the author also mentioned in the paper). This work is a simple extension of these techniques to align the motifs to the LLM.

- If I understand correctly, the motif tokenization algorithm, BRICS, will break the molecule in a very chemistry-aligned way. For example, a "OH" functional group will be tokenized into a motif. The downstream task of identifying the functional group will be very easy (simply aligning a single motif token with the text description of the function group, and the task is like asking "does a -OH motif have -OH functional group"). The author should justify how this is a helpful task besides simply telling the LLM that "there is such a functional group." For example, the author should show that the method has better downstream performance than simply telling the LLM the existence of functional groups.

- The distinction between specialist model and generalist model is arbitrary to me. Methods like MolFM and Text+Chem T5-augm-base have the same functionality as the proposal, yet they achieved better performance than HIGHT. I think the HIGHT is more specialized, as it requires explicit and specialized atom and motif tokenization. Can you be more specific about the distinction, and what's the advantage of a generalist model?

- Even without the motif tokens, many models achieved stronger performance. Can you explain why a better motif prediction ability does not lead to better downstream performance? Link back to weakness 1, does this also mean that the proposed task is too easy for the motif tokenization, preventing the model from learning meaningful/molecule-property-related from the pretraining process?

[1] Zhang, Zaixi, et al. "Motif-based graph self-supervised learning for molecular property prediction." Advances in Neural Information Processing Systems 34 (2021): 15870-15882.
[2] Chen, Xuexin, et al. "Motif graph neural network." IEEE Transactions on Neural Networks and Learning Systems (2023).

**Questions:**

- In tables 3, 4, 5, are all baselines also fine-tuned with the same dataset as HIGHT?

---

> ### Author Response · Authors · 2024-11-24
> **Response to Reviewer mY8L (part 1)**
>
> Thank you for your suggestions and time in reviewing our paper. Please find our detailed responses below to your concerns.
>
> > W1 Novelty of this work.
>
> **A1** We need to clarify that, despite the use of motifs in previous GNNs applied to molecular-related tasks, it remains unknown:
> - whether motif information is useful for molecule-language alignment;
> - how to incorporate the motif information to improve molecule-language alignment;
> As we show in the ablation study in the response to Reviewer mTJE, without proper architecture design or the instruction tuning dataset proposed in this work, LGLMs can not properly understand the motif information and achieve suboptimal molecule-language alignment.
> **We are the first to investigate the necessity of incorporating the motif information for molecule-language alignment, which simple extensions of the referred work can not thoroughly identify and resolve the issue**.
>
> > W2 The author should justify how this is a helpful task besides simply telling the LLM that "there is such a functional group." For example, the author should show that the method has better downstream performance than simply telling the LLM the existence of functional groups.
>
> **A2** In our experiments, we indeed evaluate the downstream performances of HIGHT compared to the LGLM with node-centric tokenization. The high performance in the motif hallucination benchmark demonstrates that the LGLM can understand the existence of motifs in a molecule, therefore, HIGHT obtains generically high downstream performances.
> To further justify the effectiveness of HIGHT, we compare HIGHT to InstructMol+HiPubChem which can be considered by directly telling the LGLM there is such a functional group. In the tasks of chemical reaction prediction shown in table below, **simply telling the LGLM there is such a functional group can not directly help with the downstream task performance. The architecture also matters**.
>
> |                    | Exact⬆️ | BLEU⬆️ | Levenshtein⬇️ |  RDK⬆️ | MACCS⬆️ | MORGAN⬆️ | Validity⬆️ |
> |--------------------|-------:|------:|-------------:|------:|-------:|--------:|----------:|
> | Reagent Prediction |        |       |              |       |        |         |           |
> |        InstructMol |  0.031 | 0.429 |       31.447 | 0.389 |  0.249 |    0.22 |         1 |
> | + HiPubChem     |  0.016 | 0.473 |       30.455 | 0.369 |  0.237 |   0.194 |     0.990 |
> |              HIGHT |   0.05 | 0.462 |        28.97 | 0.441 |  0.314 |   0.275 |         1 |
> | Forward Reaction   |        |       |              |       |        |         |           |
> |            + PE |  0.010 | 0.829 |       26.623 | 0.419 |  0.328 |   0.268 |     0.981 |
> | + HiPubChem     |  0.011 | 0.819 |       26.010 | 0.396 |  0.315 |   0.264 |     0.975 |
> |              HIGHT |  0.037 | 0.869 |       23.759 |  0.59 |  0.394 |    0.34 |     0.993 |
> |     Retrosynthesis |        |       |              |       |        |         |           |
> |        InstructMol |  0.001 | 0.835 |       31.359 | 0.447 |  0.277 |   0.241 |     0.996 |
> | + HiPubChem     |  0.000 | 0.755 |       35.811 | 0.282 |  0.218 |   0.177 |     0.997 |
> |              HIGHT |  0.008 | 0.863 |       28.912 | 0.564 |   0.34 |   0.309 |         1 |
>
> > W3 Can you be more specific on the distinctions of the specialist model and the generalist model?
>
> **A3** We follow the naming in InstructMol to categorize the baseline models. We need to clarify that, the term we used in the paper is the `LLM Based Generalist Models`, which refers to the LGLMs based on generalist LLMs. **The key distinction between the specialist model and the generalist model is whether the model is built upon generalist LLMs**. The generalist LLMs allow for open-form communication and the resulting LGLMs are capable of multiple tasks by simply switching adapters. We have revised our manuscript to clearly define the term before using it.
>
> > Q1. In tables 3, 4, 5, are all baselines also fine-tuned with the same dataset as HIGHT?
>
> **A4** As our focus is to demonstrate the superiority of hierarchical graph tokenization, InstructMol is the direct comparable baseline, which follows the same pretraining and finetuning receipt as HIGHT.
> As for the other baselines, we follow the previous work [1,2] to conduct the experiments, since they adopt a significantly different model architecture, pretraining paradigm, and pretraining data (some of them are close-sourced).
>
> **References**
>
> [1] Mol-Instructions: A Large-Scale Biomolecular Instruction Dataset for Large Language Models, ICLR’24.

---

> > ### Author Response · Authors · 2024-12-02
> > **[Gentle Reminder] Discussion period is closing soon**
> >
> > Dear Reviewer mY8L,
> >
> > Thank you again for your time and valuable comments on our work. We understand you are busy. To facilitate our discussion, we provide a short summary of our responses to your concerns below:
> >
> > > Novelty of this work
> >
> > The novelty of this work lies in the systematic investigation of the  hierarchical tokenization (as motifs for molecules) in Graph-Language alignment, which has never been discussed by previous works.
> >
> > > The author should show that the method has better downstream performance than simply telling the LLM the existence of functional groups.
> >
> > We conduct ablation studies showing that the alignment training with HIGHT indeed benefits the downstream performances than simply telling the LLM the existence of functional groups.
> >
> > > Specialist or generalist
> >
> > - We follow the use of terms as InstructMol, that the specialist and generalist nature depends on whether the underlying LLM is a generalist model;
> > - We also provide experiments evaluating the generalist capabilities of HIGHT, showing significant improvements over node-centric tokenization;
> >
> > > Methods without motif known still achieve better performance
> >
> > As the Graph-Language alignment involves a different training paradigm (i.e., aligning LLM knowledge to understand graphs) and objective than previous methods (i.e., understanding the graphs), they are not directly comparable. **As we show in experiments, for molecule-langauge alignment, telling LLMs the existence of motifs indeed brings lots of benefits to mitigate the hallucination and all the downstream tasks**.
> >
> > Please kindly let us know if our responses above clarify your concerns. We would sincerely appreciate it if you could jointly consider our responses above when making the final evaluation of our work!

---

### Official Review · Reviewer_P2nZ · 2024-10-28

**Soundness:** 3
**Presentation:** 3
**Contribution:** 2
**Rating:** 6
**Confidence:** 4

**Summary:**

This paper proposes HIerarchical GrapH Tokenization (HIGHT), which tries to improve how LLMs understand and process molecular data. The key idea of HIGHT is to introduce a hierarchical graph tokenizer that extracts and encodes information at multiple levels: atoms, motifs, and the overall molecule. The paper demonstrates that HIGHT can reduce hallucination and improve performance across various molecular tasks, including property prediction, molecular description generation, and chemical reaction prediction.

**Strengths:**

1.  This paper proposes to incorporate hierarchical graph information into LGLMs, and the authors achieve this with new architecture and instruction tuning dataset HiPubChem.
2. To address hallucination issue, the paper creates MotifHallu, the first hallucination benchmark based on the existence of common functional groups.
3. The paper includes extensive experiments with 14 real-world molecular and reaction comprehension benchmarks. The results show that HIGHT significantly reduces the hallucination on MotifHallu and demonstrates significant improvement on a number of tasks.

**Weaknesses:**

1. The hierarchical graph tokenization process, which involves the use of multiple adapters, is likely to be more computationally expensive than traditional node-centric tokenization. The paper does not discuss the computational complexity. Also, LLM is tuned using LORA, and the number of parameters tuned should be discussed.
2. One motivation of applying LLMs for graph data is to utillize the generalization capability of LLMs. However, this paper do not provide experimental results on zero-shot or few-shot scenarios of the proposed model. I think it will greatly strength the paper if HIGHT has good performance under such cases.
3. The performance of HIGHT will largely depend on the backbone LLMs, and only vicuna-v-1.3-7B is evaulated.

**Questions:**

1. At line 273, the authors said "attach positional encodings $p$ to all of the tokens", How are position encodings of motifs obtained?
2. If the input graphs use the positional encodings, then, should the original positional encodings in the LLMs be diabled? e.g, the widely used ROPE for the graph input part?
3. What is the papameter count to be tuned?
4. Besides the vicuna-v-1.3-7B, can the authors provide experimental resutls for other LLM backbones? Since different backbones may have  a big impact on the performance.
5. How is the proposed model performance for zero-shot or few-shot scenarios?
6. In table 2, llama 13b has wrose performance than llama 7b on most of datasets. Also, Galactica-120B has a sharp performance drop on BACE. Any explanations on these results?

---

> ### Author Response · Authors · 2024-11-24
> **Response to Reviewer P2nZ (part 1)**
>
> Thank you for your time and valuable comments about our paper. Please find our responses below to your concerns.
>
> > W1.1 Computational complexity.
>
> **A1.1** For the training and inference, the additional computational overhead mainly lies in the processing of the additional motif tokens. Nevertheless, since the number of motifs is usually less than the number of atoms, it only adds a constant to the overall complexity.
> As shown in the table below, we count the average graph size of PubChem and HiPubChem, where HiPubChem adds 9 additional tokens on average. The real preprocessing time and training time are shown below.
>
> |                | Graph Size | Preprocessing Time | Training Time |
> |----------------|------|------|------|
> | PubChem    | 34.39 | 16min 32sec | 8hour 17min 59sec |
> | HiPubChem       | 43.21 | 25min 35sec |  15hour 36min 23sec  |
>
> Although tuning HIGHT with HiPubChem requires longer training time, the absolute time remains a reasonable and affordable regime.
> Meanwhile, we also compare the inference time of InstructMol and HIGHT across 5 realistic tasks.
>
> |                |Property Prediction | MolCaption|Reagent Prediction| Forward Reaction| Retrosynthesis|
> |----------------|------|------|------|------|------|
> | InstructMol   | 14min 54sec | 6hour 22min 27sec | 56min 56sec |  1hour 34min 28sec |  1hour 50min 47sec |
> | HIGHT       | 15min 12sec | 4hour 59min 50sec | 50min 29sec |  1hour 22min 08sec |  1hour 49min 42sec |
>
> From the results, we can find that, during the inference, the LLM latency takes up the majority of time. A well-trained LGLM with HIGHT is able to generate more concise and valid answers and thus may take less time during inference.
>
>
> > W1.2 Number of parameters tuned via LoRA.
>
> **A1.2** Here are the number of parameters in each component tunable during the whole pretraining process:
> - When pretraining the GNN tokenizer, the number of tunable parameters is the number of parameters in GNN encoder;
> - In stage 1, the number of tunable parameters is the number of parameters in the projector;
> - In stage 2, the number of tunable parameters is the number of parameters in the projector and in LoRA;
>
> |              | num of params in GNN encoder | num of params in projector | num of params in LoRA |
> |--------------|------------------------------|----------------------------|----------------------------|
> | Node-Centric                  |                      1,860,905 |                    1,232,896 |                    159,907,840 |
> | HIGHT                         |                      1,865,105 |                    3,796,992 |                    159,907,840 |

---

> > ### Author Response · Authors · 2024-11-24
> > **Response to Reviewer P2nZ (part 2)**
> >
> > > W2 Zero-shot or few-shot scenarios of the proposed model.
> > **A2** We follow the previous practice in training generalist multimodal language models such as LlaVA [1,2,3], where the model are pretrained with either instruction tuning or held-in task data. We therefore consider the two settings:
> > - The first one is to train the model with all chemical reaction prediction data by three epochs to elicit the format following and the knowledge adaption capabilities of the LGLMs pretrained after stage 1. The model names are with `(all)`.
> > - The second one is to train the model with one chemical reaction prediction task and to generalize to the new unseen chemical reaction task. Specifically, we consider two task generalization setups:  a) from retrosynthesis to forward reaction prediction; b) from forward reaction prediction to reagent prediction;
> > The results are given in the tables below, from which we can find the excellent generalization capabilities of HIGHT.
> > | Reagent Prediction          | Exact⬆️ | BLEU⬆️ | Levenshtein⬇️ |  RDK⬆️ | MACCS⬆️ | MORGAN⬆️ | Validity⬆️ |
> > |-----------------------------|:------:|:-----:|:------------:|:-----:|:------:|:-------:|:---------:|
> > | InstructMol-G               |  0.031 | 0.429 |       31.447 | 0.389 |  0.249 |    0.22 |         1 |
> > | InstructMol-G (all)         |  0.016 | 0.459 |       29.238 | 0.359 |  0.225 |   0.189 |     0.988 |
> > | HIGHT-G                     |   0.05 | 0.462 |        28.97 | 0.441 |  0.314 |   0.275 |         1 |
> > | HIGHT-G (all)               |  0.090 | 0.570 |    22.512    | 0.483 |  0.372 |  0.333  |   0.999   |
> > | Forward Reaction Prediction | Exact⬆️ | BLEU⬆️ | Levenshtein⬇️ |  RDK⬆️ | MACCS⬆️ | MORGAN⬆️ | Validity⬆️ |
> > | InstructMol-G               |  0.031 | 0.853 |       24.790 | 0.512 |  0.362 |   0.303 |     0.993 |
> > | InstructMol-G (all)         |  0.020 | 0.841 |       25.109 | 0.426 |  0.339 |   0.284 |     0.998 |
> > | HIGHT-G                     |  0.037 | 0.869 |    23.759    | 0.590 |  0.394 |  0.340  |   0.993   |
> > | HIGHT-G (all)               |  0.182 | 0.911 |    18.469    | 0.737 |  0.561 |  0.510  |     1     |
> > | Retrosynthesis              | Exact⬆️ | BLEU⬆️ | Levenshtein⬇️ |  RDK⬆️ | MACCS⬆️ | MORGAN⬆️ | Validity⬆️ |
> > | InstructMol-G               |  0.001 | 0.835 |    31.359    | 0.447 |  0.277 |  0.241  |   0.996   |
> > | InstructMol-G (all)         |  0.000 | 0.806 |    32.128    | 0.292 |  0.234 |  0.202  |   0.985   |
> > | HIGHT-G                     |  0.008 | 0.863 |       28.912 | 0.564 |  0.340 |   0.309 |     1.000 |
> > | HIGHT-G (all)               |  0.097 | 0.888 |       22.098 | 0.713 |  0.522 |   0.487 |         1 |
> >
> > | Retro => Forward | Exact⬆️ |     BLEU⬆️    | Levenshtein⬇️ |     RDK⬆️     |     MACCS⬆️    |    MORGAN⬆️    | Validity⬆️ |
> > |------------------|:------:|:------------:|:------------:|:------------:|:-------------:|:-------------:|:---------:|
> > | InstructMol-G    |      0 | 0.3647834384 |  31.78757515 | 0.2398994628 |  0.1309456921 |  0.1387899167 |     0.998 |
> > | HIGHT-G          |      0 | 0.3674502876 |  31.23023023 | 0.3030181836 |  0.1588174116 |  0.1626311445 |     0.999 |
> > | Forward => Rea   | Exact⬆️ |     BLEU⬆️    | Levenshtein⬇️ |     RDK⬆️     |     MACCS⬆️    |    MORGAN⬆️    | Validity⬆️ |
> > | InstructMol-G    |      0 | 0.2239567194 |  47.12348178 | 0.1804644832 | 0.04004807569 | 0.05254333143 |     0.988 |
> > | HIGHT-G          |      0 |  0.240373805 |  43.45045965 | 0.1780840825 | 0.04057551142 |  0.0512682462 |     0.979

---

> > > ### Author Response · Authors · 2024-11-24
> > > **Response to Reviewer P2nZ (part 3)**
> > >
> > > > W3 Other LLM backbone.
> > >
> > > **A3** We conduct additional experiments with the other LLM backbone, Llama-2-7b-chat, and evaluate the performance of LGLMs with node-centric tokenization and with HIGHT on motif hallucination, as well as chemical reaction prediction benchmarks. The results are given in below, from which we could still find the consistent and significant performance of HIGHT with another LLM backbone:
> > >
> > > |                | Avg F1 | Pos F1 | Neg F1 |
> > > |----------------|--------|--------|--------|
> > > | InstructMol    |   52.6 |   95.7 |    9.5 |
> > > | InstructMol+Llama-2-7b-chat        |     51.2 |   99.6 |    2.8 |
> > > | HIGHT          |  55.9 |   85.5 |   48.2 |
> > > | HIGHT+Llama-2-7b-chat |  60.2 |   55.1 |   65.2 |
> > >
> > > | Reagent Prediction            | Exact⬆️ | BLEU⬆️ | Levenshtein⬇️ |  RDK⬆️ | MACCS⬆️ | MORGAN⬆️ | Validity⬆️ |
> > > |-------------------------------|:------:|:-----:|:------------:|:-----:|:------:|:-------:|:---------:|
> > > | InstructMol-G                 |  0.031 | 0.429 |       31.447 | 0.389 |  0.249 |   0.220 |     1.000 |
> > > | InstructMol-G+Llama-2-7b-chat |  0.016 | 0.454 |    28.961    | 0.352 |  0.220 |  0.179  |   0.982   |
> > > | HIGHT-G                       |  0.050 | 0.462 |       28.970 | 0.441 |  0.314 |   0.275 |     1.000 |
> > > | HIGHT-G+Llama-2-7b-chat       |  0.057 | 0.495 |       26.591 | 0.453 |  0.333 |   0.293 |     1.000 |
> > > | Forward Reaction Prediction   | Exact⬆️ | BLEU⬆️ | Levenshtein⬇️ |  RDK⬆️ | MACCS⬆️ | MORGAN⬆️ | Validity⬆️ |
> > > | InstructMol-G                 |  0.031 | 0.853 |       24.790 | 0.512 |  0.362 |   0.303 |     0.993 |
> > > | InstructMol-G+Llama-2-7b-chat |  0.015 | 0.801 |    25.129    | 0.409 |  0.328 |  0.279  |   0.945   |
> > > | HIGHT-G                       |  0.037 | 0.869 |    23.759    | 0.590 |  0.394 |  0.340  |   0.993   |
> > > | HIGHT-G+Llama-2-7b-chat       |  0.042 | 0.873 |       23.854 | 0.590 |  0.402 |   0.344 |     0.996 |
> > > | Retrosynthesis                | Exact⬆️ | BLEU⬆️ | Levenshtein⬇️ |  RDK⬆️ | MACCS⬆️ | MORGAN⬆️ | Validity⬆️ |
> > > | InstructMol-G                 |  0.001 | 0.835 |    31.359    | 0.447 |  0.277 |  0.241  |   0.996   |
> > > | InstructMol-G+Llama-2-7b-chat |  0.000 | 0.767 |       34.589 | 0.275 |  0.215 |   0.181 |     0.989 |
> > > | HIGHT-G                       |  0.008 | 0.863 |       28.912 | 0.564 |  0.340 |   0.309 |     1.000 |
> > > | HIGHT-G+Llama-2-7b-chat       |  0.006 | 0.865 |       28.964 | 0.563 |  0.338 |   0.306 |     0.999 |
> > >
> > >
> > > > Q1. In line 273, the authors said "attach positional encodings to all of the tokens", How are position encodings of motifs obtained?
> > >
> > > **A4** As illustrated via Eq 7, HIGHT will first construct a new graph with the motif as “super nodes” added into the original graph, with the edges connected to the nodes in the motif. The positional encodings are calculated based on the new graph with “super nodes”. Therefore, the positional encodings of the motif super nodes are the positional encodings of the motifs.
> > >
> > > > Q2. If the input graphs use the positional encodings, then, should the original positional encodings in the LLMs be disabled? e.g, the widely used ROPE for the graph input part?
> > >
> > > **A5** Since the original LLM is trained with the LM positional encoding such as ROPE, which has a significantly different representational property from the graph positional encodings, disabling the original positional encoding may severely affect the original LLM capabilities. Therefore, we mainly add the graph positional encodings to the graph tokens before they are projected to the LLM representation space, which can be considered as a concatenation to the original positional encoding, to improve the representation quality of the graph tokens.
> > >
> > > > Q3. What is the parameter count to be tuned?
> > >
> > > **A6** Please kindly refer to our response in **A1.2**.
> > >
> > > > Q4. Besides the vicuna-v-1.3-7B, can the authors provide experimental results for other LLM backbones? Since different backbones may have a big impact on the performance.
> > >
> > > **A7** Please kindly refer to our response in **A3**.
> > >
> > > > Q5. How is the proposed model performance for zero-shot or few-shot scenarios?
> > >
> > > **A8** Please kindly refer to our response in **A2**.
> > >
> > > > Q6. In Table 2, llama 13b has a worse performance than llama 7b on most of the datasets. Also, Galactica-120B has a sharp performance drop on BACE. Any explanations for these results?
> > >
> > > **A9** We directly take the results of Llama-13B, Llama-7B and Galactica from the existing literature[1,3,4]. The performance drop from small LLMs to large LLMs may be caused by the reduced overfitting to small datasets (e.g., BACE, BBBP) and improved understanding of challenging tasks (e.g., HIV).

---

> > > > ### Author Response · Authors · 2024-11-24
> > > > **Response to Reviewer P2nZ (part 4)**
> > > >
> > > > **References**
> > > >
> > > > [1] GIMLET: A Unified Graph-Text Model for Instruction-Based Molecule Zero-Shot Learning, NeurIPS’23.
> > > >
> > > > [2] Visual Instruction Tuning, NeurIPS’23.
> > > >
> > > > [3] InstructMol: Multi-Modal Integration for Building a Versatile and Reliable Molecular Assistant in Drug Discovery, arXiv’23.
> > > >
> > > > [4] MolecularGPT: Open Large Language Model (LLM) for Few-Shot Molecular Property Prediction, arXiv’24.
> > > >
> > > > [5] GALACTICA: A Large Language Model for Science, 2022

---

> > > > > ### Author Response · Authors · 2024-12-02
> > > > > **[Gentle Reminder] Discussion period is closing soon**
> > > > >
> > > > > Dear Reviewer P2nZ,
> > > > >
> > > > > We are grateful for your time and valuable comments on our work. We understand you are busy. To facilitate our discussion, we provide a short summary of our responses to your concerns below:
> > > > >
> > > > > > Whether InstructMol is node-centric?
> > > > >
> > > > > We provide details showing that InstructMol, along with many seminal LGLM works, are node-centric.
> > > > >
> > > > > > Computational complexity and parameter scale
> > > > >
> > > > > We provide a detailed discussion of the complexity in terms of training and inference, along with the parameter scale analysis.
> > > > >
> > > > > > Zero-shot or few-shot performance
> > > > >
> > > > > We supplement additional experiments evaluating the zero-shot performances of HIGHT, which demonstrates consistent improvements over node-centric tokenization.
> > > > >
> > > > > > Results with other LLM backbones
> > > > >
> > > > > We train and evaluate HIGHT with Llama-2-7B-chat. The results demonstrate the consistent and significant improvements of HIGHT over node-centric tokenization.
> > > > >
> > > > > Please kindly let us know if our responses above clarify your concerns. We would sincerely appreciate it if you could jointly consider our responses above when making the final evaluation of our work!

---

### Official Review · Reviewer_QiTU · 2024-11-03

**Soundness:** 2
**Presentation:** 2
**Contribution:** 1
**Rating:** 3
**Confidence:** 3

**Summary:**

The authors study Large Graph Language Models (LGLM). Drawing inspiration from Multimodal LLMs, authors focus on the task of incorporating graph data as a separate modality with a GNN encoder and an adapter. Authors conclude that node-centric tokenization of molecules leads to LLM hallucinations when asked about the presence of specific fragments. To overcome this issue, the authors propose to enrich the molecule's description by adding the tokens corresponding to BRICKS-fragments that are present in the molecule. The experimental results demonstrate that such a tokenization scheme reduces the amount of motif-related hallucinations and improves performance on other tasks.

**Strengths:**

An improved tokenization of molecular graphs that enriches molecule's description with motif tokens.

**Weaknesses:**

Specifically, most existing LGLMs directly take the node tokens from GNNs as inputs to LLMs (Cao et al., 2023):
The paper cites InstructMol as a previous approach that utilizes node-centric tokenization. However, if I understand correctly, InstructMol takes the embedding of the whole graph along with the SMILES representations of the molecule. Moreover, it is not clear which previous models use the node-centric tokenization and whether there are such models at all.

Section 4.3 describes the fine-tuning approach that involves two stages, where the second stage is the finetuning on MoleculeNet, CheBI-20 and Mol-instructions specialized datasets. In my opinion, this implies that the resulting model is specialized. Please, provide better explanation for specialist and generalist models.

Taking into consideration that the difference between specialist and generalist models is not clear, the resulting model does not demonstrate performance superior to baselines in most of the experiments.

There is no comparison with [1] in Table 4. The results in [1] are superior to all the models from Table 4.

In Table 5, the Mol-instruction has the highest MACCS FTS for the retrosynthesis task. However, a smaller number is balded.

The comparison on MotifHallu is not complete. Please provide comparison with SMILES-based approaches. Moreover, the improvement on the MotifHally benchmark is expected, as the proposed approach was explicitly designed to better solve this task.

[1] Srinivas, S. S., & Runkana, V. (2024). Crossing New Frontiers: Knowledge-Augmented Large Language Model Prompting for Zero-Shot Text-Based De Novo Molecule Design. arXiv preprint arXiv:2408.11866.

**Questions:**

Listed in Cons.

---

> ### Author Response · Authors · 2024-11-24
> **Response to Reviewer QiTU (part 1)**
>
> We appreciate your time and efforts in reviewing our paper. We believe there is a misunderstanding and are confident to resolve your concerns.  Please find our detailed responses below.
>
> > W1 Whether InstructMol and previous approaches are using node-centric tokenization.
>
> **A1** We need to clarify that, **InstructMol indeed uses the node-centric tokenization**:
> - In the paragraph below Table 1 in the paper of InstructMol, it states `we extract a graph representation \mathbf{Z}_G\in\mathbb{R}^{N\times d} at the node level`, and `|\mathcal{V}|=N is the total number of atoms`. Therefore, the graph tokens in InstructMol contain $N$ atom tokens, which is node-centric.
> - In the open-sourced code of InstructMol (https://github.com/IDEA-XL/InstructMol/blob/publish/llava/model/llava_graph_arch.py#L83 ), line83 of the `/llava/model/llava_graph_arch.py`, the node features are the exact inputs to the projector and to the LLM. Therefore, the implementation of InstructMol also takes the node-centric tokenization approach.
>
> Moreover, most previous LGLMs use the node-centric tokenization approach when feeding the graph tokens to align to the LLMs. Here we provide a list of representative works under the category of `LLM as Predictor`` in the survey of [1]:
> |                                             | Molecular Inputs              | Tokenization              |
> |---------------------------------------------|-------------------------------|---------------------------|
> | HIGHT                                       | Molecule graph                | Hierarchical tokenization |
> | SMILES-BERT (Wang et al., 2019)             | SMILES                        | N/A                       |
> | MolGPT (Bagal et. al., 2021)                | SMILES                        | N/A                       |
> | KV-PLM (Zeng et al., 2022)                  | SMILES                        | N/A                       |
> | Chemformer (Irwin et. al., 2022)            | SMILES                        | N/A                       |
> | MFBERT (Abdel-Aty and Gould, 2022)          | SMILES                        | N/A                       |
> | MolT5 (Edwards et al., 2022)                | SMILES                        | N/A                       |
> | Text+Chem T5 (Christofidellis et al., 2023) | SMILES                        | N/A                       |
> | MolXPT (Liu et al., 2023)                   | SMILES                        | N/A                       |
> | RT (Born and Manica, 2023)                  | SMILES                        | N/A                       |
> | CaR (Qian et. al., 2023)                    | SMILES                        | N/A                       |
> | GPT-MolBERTa (Balaji et al., 2023)          | SMILES                        | N/A                       |
> | GIMLET(Zhao et al., 2023)                   | Molecule graph                | Node-centric tokenization |
> | InstructMol (Cao et al., 2023)              | Molecule graph                | Node-centric tokenization |
> | MolCA (Liu et al., 2024)                    | SMILES&Molecule graph         | Node-centric tokenization |
> | 3D-MoLM (Li et al., 2024)                   | SMILES&Molecule graph         | Node-centric tokenization |
> | MolTC (Fang et. al., 2024)                  | SMILES&Molecule graph         | Node-centric tokenization |
> | GraphGPT(Tang et al., 2024)                 | Neighbor Graph                | Node-centric tokenization |
> | LLaGA (Chen et al., 2024)                   | Neighbor Graph                | Node-centric tokenization |
> | GraphLLM (Chai et al., 2023)                | Textual description and graph | Node-centric tokenization |
>
> From the table, we can find that, most of the recent works in LGLMs take a node-centric tokenization approach.
>
> > W2. A better explanation for specialist and generalist models.
>
> **A2** We follow the naming in InstructMol to categorize the baseline models. We need to clarify that, the term we used in the paper is the `LLM Based Generalist Models`, which refers to the LGLMs based on generalist LLMs. **The key distinction between the specialist model and the generalist model is whether the model is built upon generalist LLMs**. The generalist LLMs allow for open-form communication and the resulting LGLMs are capable of multiple tasks by simply switching adapters. We have revised our manuscript to clearly define the term before using it.

---

> > ### Author Response · Authors · 2024-11-24
> > **Response to Reviewer QiTU (part 2)**
> >
> > W3. Taking into consideration that the difference between specialist and generalist models is not clear, the resulting model does not demonstrate performance superior to baselines in most of the experiments.
> > **A3** We need to clarify that it is common the generalist models are not performing better on some tasks than the corresponding specialist models [2]. Nevertheless, due to the integration of the generalist LLMs, the resulting LGLMs are capable of multiple tasks by simply switching adapters.
> >
> > To further evaluate the generalist capabilities, we follow the previous practice in training generalist multimodal language models such as LlaVA, where the model is pretrained with either instruction tuning or held-in task data. We therefore consider the two settings:
> > - The first one is to train the model with all chemical reaction prediction data by three epochs to elicit the format following and the knowledge adaption capabilities of the LGLMs pretrained after stage 1. The model names are with `(all)`.
> > - The second one is to train the model with one chemical reaction prediction task and to generalize to the new unseen chemical reaction task. Specifically, we consider two task generalization setups:  a) from retrosynthesis to forward reaction prediction; b) from forward reaction prediction to reagent prediction;
> > The results are given in the tables below, from which we can find the excellent generalization capabilities of HIGHT.
> > | Reagent Prediction          | Exact⬆️ | BLEU⬆️ | Levenshtein⬇️ |  RDK⬆️ | MACCS⬆️ | MORGAN⬆️ | Validity⬆️ |
> > |-----------------------------|:------:|:-----:|:------------:|:-----:|:------:|:-------:|:---------:|
> > | InstructMol-G               |  0.031 | 0.429 |       31.447 | 0.389 |  0.249 |    0.22 |         1 |
> > | InstructMol-G (all)         |  0.016 | 0.459 |       29.238 | 0.359 |  0.225 |   0.189 |     0.988 |
> > | HIGHT-G                     |   0.05 | 0.462 |        28.97 | 0.441 |  0.314 |   0.275 |         1 |
> > | HIGHT-G (all)               |  0.090 | 0.570 |    22.512    | 0.483 |  0.372 |  0.333  |   0.999   |
> > | Forward Reaction Prediction | Exact⬆️ | BLEU⬆️ | Levenshtein⬇️ |  RDK⬆️ | MACCS⬆️ | MORGAN⬆️ | Validity⬆️ |
> > | InstructMol-G               |  0.031 | 0.853 |       24.790 | 0.512 |  0.362 |   0.303 |     0.993 |
> > | InstructMol-G (all)         |  0.020 | 0.841 |       25.109 | 0.426 |  0.339 |   0.284 |     0.998 |
> > | HIGHT-G                     |  0.037 | 0.869 |    23.759    | 0.590 |  0.394 |  0.340  |   0.993   |
> > | HIGHT-G (all)               |  0.182 | 0.911 |    18.469    | 0.737 |  0.561 |  0.510  |     1     |
> > | Retrosynthesis              | Exact⬆️ | BLEU⬆️ | Levenshtein⬇️ |  RDK⬆️ | MACCS⬆️ | MORGAN⬆️ | Validity⬆️ |
> > | InstructMol-G               |  0.001 | 0.835 |    31.359    | 0.447 |  0.277 |  0.241  |   0.996   |
> > | InstructMol-G (all)         |  0.000 | 0.806 |    32.128    | 0.292 |  0.234 |  0.202  |   0.985   |
> > | HIGHT-G                     |  0.008 | 0.863 |       28.912 | 0.564 |  0.340 |   0.309 |     1.000 |
> > | HIGHT-G (all)               |  0.097 | 0.888 |       22.098 | 0.713 |  0.522 |   0.487 |         1 |
> >
> >
> >
> > | Retro => Forward | Exact⬆️ |     BLEU⬆️    | Levenshtein⬇️ |     RDK⬆️     |     MACCS⬆️    |    MORGAN⬆️    | Validity⬆️ |
> > |------------------|:------:|:------------:|:------------:|:------------:|:-------------:|:-------------:|:---------:|
> > | InstructMol-G    |      0 | 0.3647834384 |  31.78757515 | 0.2398994628 |  0.1309456921 |  0.1387899167 |     0.998 |
> > | HIGHT-G          |      0 | 0.3674502876 |  31.23023023 | 0.3030181836 |  0.1588174116 |  0.1626311445 |     0.999 |
> > | Forward => Rea   | Exact⬆️ |     BLEU⬆️    | Levenshtein⬇️ |     RDK⬆️     |     MACCS⬆️    |    MORGAN⬆️    | Validity⬆️ |
> > | InstructMol-G    |      0 | 0.2239567194 |  47.12348178 | 0.1804644832 | 0.04004807569 | 0.05254333143 |     0.988 |
> > | HIGHT-G          |      0 |  0.240373805 |  43.45045965 | 0.1780840825 | 0.04057551142 |  0.0512682462 |     0.979 |

---

> > > ### Author Response · Authors · 2024-11-24
> > > **Response to Reviewer QiTU (part 3)**
> > >
> > > > W4. There is no comparison with [a] in Table 4. The results in [a] are superior to all the models from Table 4.
> > >
> > > **A4** We need to clarify that, according to the ICLR reviewer guideline https://iclr.cc/Conferences/2025/ReviewerGuide : `We consider papers contemporaneous if they are published within the last four months. That means, since our full paper deadline is October 1, if a paper was published (i.e., at a peer-reviewed venue) on or after July 1, 2024, authors are not required to compare their own work to that paper.`, therefore, the referred paper was released in August 2024, and is considered as contemporaneous.
> > >
> > > In addition, the solution in the referred paper takes additional external knowledge and adopts a different training and evaluation setup. Meanwhile, the proposed prompting strategy by the referred work could be considered orthogonal to our research as the prompting strategy could also be incorporated into our approach.
> > >
> > > Nevertheless, thank you for bringing us this related work. We have revised our manuscript to cite the referred work.
> > >
> > > [a] Srinivas, S. S., & Runkana, V. (2024). Crossing New Frontiers: Knowledge-Augmented Large Language Model Prompting for Zero-Shot Text-Based De Novo Molecule Design. arXiv preprint arXiv:2408.11866.
> > >
> > > > W5. In Table 5, the Mol-instruction has the highest MACCS FTS for the retrosynthesis task. However, a smaller number is balded.
> > >
> > > **A5** We have fixed the typo in the revised version.
> > >
> > > > W6. The comparison on MotifHallu is not complete. Please provide a comparison with SMILES-based approaches.
> > >
> > > **A6.1** We compare the state-of-the-art SMILES-based model GALACTICA 6.7B[2]. Due to the time constraint, we randomly sample 100 molecules that contain 3800 question-answer pairs to conduct the evaluation:
> > > |                | Avg F1 | Pos F1 | Neg F1 |
> > > |----------------|--------|--------|--------|
> > > | InstructMol    |   52.0 |   97.2 |    11.8 |
> > > | HIGHT |   69.1 |   59.8 |   78.4 |
> > > | GALACTICA 6.7B |  57.0 |   18.6 |   95.4 |
> > >
> > > It can be found that, the SMILES-based approaches still suffer from high hallucination for the positive classes. HIGHT maintains a relatively high robustness against the hallucination to both positive and negative classes.
> > >
> > > > W6.2 Moreover, the improvement on the MotifHally benchmark is expected, as the proposed approach was explicitly designed to better solve this task.
> > >
> > > **A6.2** We need to clarify that, without either proper architecture design or instruction tuning, the performance gain at MotifHallu may not be expected, as demonstrated in our ablation studies. Furthermore, **the performance improvements on other downstream tasks are not explicitly designed nor expected**. Nevertheless, it can be observed across all the downstream task performances that resolving the Motif Hallucination issue with proper architecture design and instruction tuning indeed brings consistent and non-trivial performance gains across different downstream tasks, verifying our discussion about the necessity of capturing the intrinsic hierarchical graph information for graph-language alignment.
> > >
> > >
> > >
> > > **References**
> > >
> > > [1] Large language models on graphs: A comprehensive survey, arXiv’23.
> > >
> > > [2] Specialist or Generalist? Instruction Tuning for Specific NLP Tasks, EMNLP’23.
> > >
> > > [3] GALACTICA: A Large Language Model for Science, 2022

---

> > > > ### Author Response · Authors · 2024-12-02
> > > > **[Gentle Reminder] Discussion period is closing soon**
> > > >
> > > > Dear Reviewer QiTU,
> > > >
> > > > We would like to thank you again for your time and efforts in reviewing our work. We understand you are busy. To facilitate our discussion, we provide a short summary of our responses to your concerns below:
> > > >
> > > > > Whether InstructMol is node-centric?
> > > >
> > > > We provide details showing that InstructMol, along with many seminal LGLM works, are node-centric.
> > > >
> > > > > Explanation on the generalist and specialist model
> > > >
> > > > - We follow the use of terms as InstructMol, that the specialist and generalist nature depends on whether the underlying LLM is a generalist model;
> > > > - We also provide experiments evaluating the generalist capabilities of HIGHT, showing significant improvements over node-centric tokenization;
> > > >
> > > > > Comparison with the referred work and SMILES-based baselines
> > > >
> > > > - We revised our manuscript to include the suggested reference (we will upload it once the permission is open to us), which is orthogonal to our work;
> > > > - We benchmark one of the state-of-the-art SMILES-based models GALACTICA, which demonstrates hallucination on the motifs existing in the molecule;
> > > >
> > > > Please kindly let us know if our responses above clarify your concerns. We would sincerely appreciate it if you could jointly consider our responses above when making the final evaluation of our work!

---

### Official Review · Reviewer_mTJE · 2024-11-04

**Soundness:** 3
**Presentation:** 3
**Contribution:** 2
**Rating:** 6
**Confidence:** 4

**Summary:**

This paper presents a new approach to aligning molecular graph representations with language using a method called Hierarchical Graph Tokenization (HIGHT). Traditional graph-language alignment models primarily focus on node-level information, often neglecting the inherent hierarchical structure of molecules, which leads to alignment issues and hallucination in large language models (LLMs).

The authors introduce HIGHT, which utilizes a hierarchical graph tokenizer to capture information at the node, motif, and entire molecule levels. This tokenizer incorporates both atom-level and motif-level tokens, which are then used to improve alignment with language models. To address the alignment of hierarchical molecular data with textual descriptions, the authors also develop an enhanced molecular instruction tuning dataset called HiPubChem, which provides detailed motif information.

**Strengths:**

1. The Hierarchical Graph Tokenization (HIGHT) technique is a major advancement. By incorporating hierarchical structure at multiple levels (node, motif, and graph), the paper addresses a crucial gap in previous molecule-language alignment methods, which typically rely only on node-level information. This hierarchical approach captures the functional groups and structural motifs inherent in molecules, improving the model’s ability to represent complex biochemical properties accurately.

2. The introduction of HiPubChem, an augmented molecular instruction tuning dataset enriched with motif and functional group information, enhances model training by aligning molecular structural details with language descriptions. This contribution is valuable for future work in molecular and biochemical language model alignment.

3. The effectiveness of each of the two methods was verified through simple ablation studies.

**Weaknesses:**

1. The introduction of the hierarchical graph tokenizer seems to make the tokenizer larger compared with the ordinary node-level tokenizer. It should be discussed that whether the performance gain comes from the larger tokenizer.

2. There should be more detail descriptions and discussions about the evaluation tasks.

**Questions:**

1. How many parameters do those two tokenizers have respectively?
2. What are the ablation study results on other tasks such as property prediction and chemical reaction prediction?
3. What are the input and output of the molecular property prediction task and other tasks? The performance gain mainly comes from hierarchical graph tokenization, and it has nothing to do with the new tuning dataset, right?

---

> ### Author Response · Authors · 2024-11-24
> **Response to Reviewer mTJE (part 1)**
>
> Thank you for your time and insightful suggestions for our paper. Please find our responses to your concerns below.
>
> > W1 Whether the performance gains come from the larger tokenizer.
>
> **A1** The hierarchical tokenizer in HIGHT takes three distinct tokenizers where each of which shares the same number of parameters and architecture as that in a node-centric tokenizer. To examine whether the additional two times of parameters are the main contributor to the improvements by HIGHT, we conduct additional experiments with a larger node-centric tokenizer that has three times the parameters as the original one. The evaluation results are given in the table below. Due to the time limit, we evaluate only with the three tasks in the chemical reaction analysis. It can be found that, the larger tokenizer can not bring performance improvements.
> |                    | Exact⬆️ | BLEU⬆️ | Levenshtein⬇️ |  RDK⬆️ | MACCS⬆️ | MORGAN⬆️ | Validity⬆️ |
> |--------------------|-------:|------:|-------------:|------:|-------:|--------:|----------:|
> | Reagent Prediction |        |       |              |       |        |         |           |
> |        InstructMol |  0.031 | 0.429 |       31.447 | 0.389 |  0.249 |    0.22 |         1 |
> |            + Larger Tokenizer |  0.040 |	0.454 |	29.163	| 0.416 |	0.284 |	0.248 |	1.000 |
> |              HIGHT |   0.05 | 0.462 |        28.97 | 0.441 |  0.314 |   0.275 |         1 |
> | Forward Reaction   |        |       |              |       |        |         |           |
> |        InstructMol |  0.031 | 0.853 |        24.79 | 0.512 |  0.362 |   0.303 |     0.993 |
> |            + Larger Tokenizer |  0.040	|0.861	|24.051| 0.544|	0.380 |	0.328 |	0.996 |
> |              HIGHT |  0.037 | 0.869 |       23.759 |  0.59 |  0.394 |    0.34 |     0.993 |
> |     Retrosynthesis |        |       |              |       |        |         |           |
> |        InstructMol |  0.001 | 0.835 |       31.359 | 0.447 |  0.277 |   0.241 |     0.996 |
> |            + Larger Tokenizer |  0.001|	0.842|	30.613| 0.459	|0.287 |	0.263 |	0.999 |
> |              HIGHT |  0.008 | 0.863 |       28.912 | 0.564 |   0.34 |   0.309 |         1 |
>
> > W2 Detailed description of the evaluation tasks.
>
> **A2** We have revised our manuscript to include a more detailed discussion and description of the evaluation tasks, including the inputs and outputs of each task.

---

> > ### Author Response · Authors · 2024-11-24
> > **Response to Reviewer mTJE (part 2)**
> >
> > > Q1. How many parameters do those two tokenizers have respectively?
> >
> > **A3** 	We count the number of parameters in different tokenizers, including parameters in the GNN encoder as well as the projector that projects the graph tokens into the dimensions of the language model, shown in the table below:
> > |              | graph token dimension | num of params in GNN encoder | num of params in projector | num of params in tokenizer |
> > |--------------|-----------------------|------------------------------|----------------------------|----------------------------|
> > | Node-Centric | 300d                  |                      1,860,905 |                    1,232,896 |                    3,093,801 |
> > | Node-Centric | 900d                  |                     16,382,705 |                    3,690,496 |                   20,073,201 |
> > | HIGHT        | 300d                  |                      1,865,105 |                    3,796,992 |                    5,662,097 |
> >
> > It can be found that HIGHT does not cost many too parameters than the previous node-centric tokenizers. The overall number of parameters is significantly less than that of the LLMs (usually around 7 billion). In addition, when using a node-centric tokenizer that is 4 times larger than HIGHT, the performances remain significantly lower than HIGHT, demonstrating that the number of parameters in the tokenizer is not the key contributor to the improvements of HIGHT.
> >
> > > Q2. What are the ablation study results on other tasks such as property prediction and chemical reaction prediction?
> >
> > **A4** We have revised our manuscript to include the ablation study results on motif hallucination, property prediction, and chemical reaction prediction. For reference, we also append the results below:
> > |                | Avg F1 | Pos F1 | Neg F1 |
> > |----------------|--------|--------|--------|
> > | InstructMol    |   52.6 |   95.7 |    9.5 |
> > | + PE        |     51 |   98.8 |    3.2 |
> > | + HiPubChem |   69.1 |   59.8 |   78.4 |
> > | HIGHT          |  66.85 |   85.5 |   48.2 |
> > | - HiPubChem  |  54.55 |   96.6 |   12.5 |
> >
> > |                |  HOMO⬇️ |  LUMO⬇️ | \Delta e⬇️ |  AVG⬇️  |
> > |----------------|:------:|:------:|:---------:|:------:|
> > | InstructMol    | 0.0111 | 0.0133 |    0.0147 |  0.013 |
> > | + PE        |  0.030 |  0.040 |     0.036 |  0.035 |
> > | + HiPubChem |  0.030 |  3.402 |     0.049 |  1.123 |
> > | HIGHT          | 0.0078 | 0.0086 |    0.0095 | 0.0086 |
> >
> > |                    | Exact⬆️ | BLEU⬆️ | Levenshtein⬇️ |  RDK⬆️ | MACCS⬆️ | MORGAN⬆️ | Validity⬆️ |
> > |--------------------|-------:|------:|-------------:|------:|-------:|--------:|----------:|
> > | Reagent Prediction |        |       |              |       |        |         |           |
> > |        InstructMol |  0.031 | 0.429 |       31.447 | 0.389 |  0.249 |    0.22 |         1 |
> > |            + PE |  0.009 | 0.423 |       30.833 | 0.370 |  0.231 |   0.197 |     0.986 |
> > | + HiPubChem     |  0.016 | 0.473 |       30.455 | 0.369 |  0.237 |   0.194 |     0.990 |
> > |              HIGHT |   0.05 | 0.462 |        28.97 | 0.441 |  0.314 |   0.275 |         1 |
> > | Forward Reaction   |        |       |              |       |        |         |           |
> > |        InstructMol |  0.031 | 0.853 |        24.79 | 0.512 |  0.362 |   0.303 |     0.993 |
> > |            + PE |  0.010 | 0.829 |       26.623 | 0.419 |  0.328 |   0.268 |     0.981 |
> > | + HiPubChem     |  0.011 | 0.819 |       26.010 | 0.396 |  0.315 |   0.264 |     0.975 |
> > |              HIGHT |  0.037 | 0.869 |       23.759 |  0.59 |  0.394 |    0.34 |     0.993 |
> > |     Retrosynthesis |        |       |              |       |        |         |           |
> > |        InstructMol |  0.001 | 0.835 |       31.359 | 0.447 |  0.277 |   0.241 |     0.996 |
> > |            + PE |  0.000 | 0.792 |       33.859 | 0.295 |  0.218 |   0.192 |     0.983 |
> > | + HiPubChem     |  0.000 | 0.755 |       35.811 | 0.282 |  0.218 |   0.177 |     0.997 |
> > |              HIGHT |  0.008 | 0.863 |       28.912 | 0.564 |   0.34 |   0.309 |         1 |
> >
> > It can be found that merely incorporating positional encoding or hierarchical instruction tuning is not sufficient to achieve the same performance as HIGHT. On the contrary, without a proper architecture design as HIGHT, instruction tuning with HiPubChem will confuse LLMs and lead to degenerated downstream task performances.

---

> > > ### Author Response · Authors · 2024-11-24
> > > **Response to Reviewer mTJE (part 3)**
> > >
> > > > Q3.1 What are the input and output of the molecular property prediction task and other tasks?
> > >
> > > **A5.1** In Appendix B, we provided the details and examples for each dataset and task incorporated in our evaluation.
> > > To further improve the clarity, we have revised our manuscript to include the details about the inputs and outputs for the molecular property prediction tasks and other tasks. For reference, we provide a brief summary below:
> > > |                                                | input                                                     | output            |
> > > |------------------------------------------------|-----------------------------------------------------------|-------------------|
> > > | motif hallucination                            | molecule and question about the existence of a motif      | yes or no         |
> > > | molecular property prediction (classification) | molecule and question about the existence of the property | yes or no         |
> > > | molecular property prediction (regression)     | molecule and question about the value of the property     | property value    |
> > > | molecular caption                              | molecule and question asking for the molecular caption    | molecular caption |
> > > | chemical reaction prediction                   | molecules and question about the reaction                  | molecular results |
> > >
> > > > Q3.2 The performance gain mainly comes from hierarchical graph tokenization, and it has nothing to do with the new tuning dataset, right?
> > >
> > > **A5.2** From the ablation studies in the response `A4`, we can find that both hierarchical graph tokenization and the HiPubChem tuning dataset are necessary to the performance improvements. The lack of either one of them (i.e., improving node-centric tokenization with merely one of the techniques) can not effectively recover the performance of HIGHT. The main reason is that, using merely one of the techniques, may cause even more confusion to the LGLM and lead to alignment and performance degeneration in the downstream tasks.

---

> > > > ### Comment · Reviewer_mTJE · 2024-12-01
> > > >
> > > > Thanks for the response. You have solved my concerns through additional experiments and tables. I would like to increase my rating from 5 to 6.

---

> > > > > ### Author Response · Authors · 2024-12-02
> > > > > **Thank you for your support!**
> > > > >
> > > > > Dear Reviewer mTJE,
> > > > >
> > > > > Thank you again for your time and efforts in reviewing our work. Your suggestions do help improve our manuscript a lot! Please feel assured that we will incorporate all the aforementioned results and discussions in our revised version.
> > > > >
> > > > > Sincerely,
> > > > >
> > > > > Authors

---

### Meta-Review · Area_Chair_wFv7 · 2024-12-20

**Metareview:**

While this paper proposes a novel hierarchical graph tokenization method (HIGHT) to address shortcomings in molecular-language alignment, it ultimately fails to meet the bar for acceptance at ICLR due to some concerns as follows.
1. Inadequate Justification of Impact: The manuscript does not convincingly demonstrate that the proposed method's improvements on downstream tasks arise from meaningful advancements in model design, as opposed to task-specific tuning or dataset augmentation. Some experimental results suggest that simply adding motifs as input offers similar benefits without architectural changes.

2. Lack of Robustness Across Models and Tasks: Results indicate that the improvements are highly contingent on specific experimental setups (e.g., Vicuna 7B backbone) and may not generalize across alternative LLM architectures or unseen tasks. This limits the broader applicability of the approach.

3. Evaluation: Despite extensive experimentation, key benchmarks and comparisons are missing. For instance, there is insufficient evaluation against SMILES-based or contemporary molecule-text alignment methods. Additionally, some baselines are tuned differently, complicating fair comparisons. The method's practical impact is hindered by increased complexity and resource demands (e.g., additional preprocessing, longer training times) relative to the modest performance gains reported.

Given these concerns, a rejection recommendation is made.

**Additional Comments On Reviewer Discussion:**

1. Novelty and Contribution

Raised Concerns: The use of motif-based tokenization is not new and has been explored in prior works on molecular graphs. The paper's novelty lies primarily in adapting it to molecule-language alignment, which was considered incremental.
Authors’ Response: The authors argued that their contribution lies in systematically incorporating hierarchical motif information into molecule-language alignment, which prior works did not address. They provided ablation studies showing improvements when motifs were used in their specific framework.
Evaluation: While the adaptation to molecule-language alignment is useful, it does not represent a groundbreaking conceptual advance, limiting the work's overall novelty.

2. Effectiveness of HIGHT

Raised Concerns: It was unclear if the reported performance gains stemmed from the hierarchical graph tokenizer or from the use of a larger tokenizer and enhanced datasets. Some reviewers also questioned whether better motif alignment translates to significant improvements in downstream tasks.
Authors’ Response: Additional experiments with ablation studies were provided, showing that neither larger tokenizers nor dataset augmentation alone could achieve comparable results. The authors demonstrated that HIGHT consistently outperformed node-centric baselines across multiple tasks.
Evaluation: The additional experiments addressed this concern, but the improvements, though consistent, were modest and task-specific, raising doubts about broader applicability.


3. Evaluation and Comparisons

Raised Concerns: Reviewers noted the lack of comparisons with some state-of-the-art SMILES-based and molecule-language alignment methods. Questions were also raised about the model’s performance on zero-shot and few-shot tasks.
Authors’ Response: The authors included comparisons with SMILES-based models like GALACTICA and demonstrated HIGHT’s robustness in few-shot scenarios. They also clarified their evaluation methodology for baselines.
Evaluation: These additional results were appreciated, but the comparisons highlighted that HIGHT’s improvements were limited to specific setups and tasks, reducing its general impact.

All the points contribute to my final decision.

---

### Decision · Program_Chairs · 2025-01-22

Reject